# CMTr cap-adjacent 2′-O-ribose mRNA methyltransferases are required for reward learning and mRNA localization to synapses

Irmgard U. Haussmann[1,2], Yanying Wu[3], Mohanakarthik P. Nallasivan[1], Nathan Archer [4], Zsuzsanna Bodi [5], Daniel Hebenstreit[6], Scott Waddell [3], Rupert Fray [5] & Matthias Soller [1,7 ✉]

Cap-adjacent nucleotides of animal, protist and viral mRNAs can be *O*-methylated at the 2′ position of the ribose (cOMe). The functions of cOMe in animals, however, remain largely unknown. Here we show that the two cap methyltransferases (CMTr1 and CMTr2) of *Drosophila* can methylate the ribose of the first nucleotide in mRNA. Double-mutant flies lack cOMe but are viable. Consistent with prominent neuronal expression, they have a reward learning defect that can be rescued by conditional expression in mushroom body neurons before training. Among CMTr targets are cell adhesion and signaling molecules. Many are relevant for learning, and are also targets of Fragile X Mental Retardation Protein (FMRP). Like FMRP, cOMe is required for localization of untranslated mRNAs to synapses and enhances binding of the cap binding complex in the nucleus. Hence, our study reveals a mechanism to co-transcriptionally prime mRNAs by cOMe for localized protein synthesis at synapses.

[1] School of Biosciences, College of Life and Environmental Sciences, University of Birmingham, Edgbaston, Birmingham B15 2TT, UK. [2] Department of Life Science, Faculty of Health, Education and Life Sciences, Birmingham City University, Birmingham B15 3TN, UK. [3] Centre for Neuronal Circuits and Behaviour, The University of Oxford, Oxford OX1 3TA, UK. [4] School of Veterinary Medicine and Sciences, University of Nottingham, Sutton Bonington, Loughborough LE12 5RD, UK. [5] School of Biosciences, Plant Science Division, University of Nottingham, Sutton Bonington, Loughborough LE12 5RD, UK. [6] School of Life Sciences, University of Warwick, Coventry CV4 7AL, UK. [7] Birmingham Centre for Genome Biology, University of Birmingham, Edgbaston, Birmingham B15 2TT, UK. ✉email: m.soller@bham.ac.uk

Methylation of cap-adjacent or internal nucleotides in messenger RNA (mRNA) is a major post-transcriptional mechanism to regulate gene expression. Methylation of mRNA is particularly prominent in the brain, but the molecular function of methylated nucleotides and their biological roles are poorly understood[1–5].

Methylation of cap-adjacent nucleotides is an abundant modification of animal, protist, and viral mRNAs, that varies in different tissues and transcripts[6–17]. The most common methylation of cap-adjacent nucleotides is O-methylation at the 2′ position of the ribose (cOMe). This modification is introduced co-transcriptionally by two dedicated cap methyltransferases (CMTr1 and CMTr2) after capping at the beginning of an mRNA to a characteristic 5′–5′ linked N7-methylated guanosine[18–20]. Knock-out of CMTr1 leads to neurological defects in mice and is embryonic lethal, while Drosophila CMTr1 null is viable, but has minor defects in siRNA-mediated gene silencing. It has been postulated that CMTr1 methylates the first and CMTr2 the second nucleotide in humans[6,20], while in trypanosomes the three CMTrs methylate the first four nucleotides[21], but the unequivocal determination of cOMe on other than the first position remains technically challenging[11,22,23].

In vertebrates, if the first nucleotide is adenosine it can also be methylated at the N6 position by PcifI, but the mechanism for cap adenosine N6-methylation is different from internal methylation of adenosine and requires the prior cOMe modification[9–13,17,24–26].

The main function of the cap is to protect mRNAs from degradation and to recruit translation initiation factors, but also to promote splicing and 3′ end processing[27]. The cap is initially bound in the nucleus by the cap-binding complex (CBC), consisting of CBP20 and CBP80. Upon export from the nucleus, CBC is replaced by eIF4E, which is predominantly cytoplasmic and rate-limiting for translation initiation[28,29]. N7-methylation of the cap guanosine is critical for both CBC and eIF4E binding. The importance of cap-adjacent nucleotide methylation in animal gene expression, however, remains elusive but is known to be essential in trypanosomes and viruses including SARS-CoV-2 for propagation[15].

## Results

***CMTrs act redundantly.*** To elucidate the biological function of cap-adjacent 2′-O-ribose methylation (cOMe) in animals we made null mutants of the CMTr1 (CG6379) and CMTr2 (adrift) genes in Drosophila, that are corresponding homologs of human CMTr1 and CMTr2 (Supplementary Fig. 1). We generated small intragenic deletions in each gene by imprecise excision of a P-element transposon to make $CMTr1^{13A}$ and $CMTr2^{M32}$ mutant flies (Fig. 1a–c). Both of these genetic lesions remove the catalytic methyltransferase domain from the encoded CMTr1 and CMTr2 proteins. Perhaps surprising, these mutant flies are viable and fertile as single and double mutants, exhibiting a slightly reduced survival to adulthood after hatching from the egg (Fig. 1d), and reduced climbing activity in negative geotaxis assays (Fig. 1e). In addition, CMTr1 mutants, and to a greater extent CMTr2 mutants, have reduced numbers of synapses at neuromuscular junctions (NMJs) of third instar larvae (Fig. 1f).

To detect cOMe in purified mRNAs we replaced the cap guanosine with a $^{32}$P-alphaGTP by first decapping mRNAs with yDcpS that leaves a di-phosphate at the first nucleotide, which is the substrate for vaccinia capping enzyme (Supplementary Fig. 2a). Digestion of such labeled mRNA with RNAse I, which is unable to cleave after 2′-O-ribose methylated nucleotides result in unmethylated $^{m7}$GpppA di-nucleotide, and 2′-O-ribose methylated tri- and tetra nucleotides that can be analyzed on 20% denaturing acrylamide gels (Fig. 1g). In adult flies, about 80% of mRNAs carry cOMe on the first nucleotide, but we could not detect methylation of the second nucleotide as compared to a single nucleotide ladder and appropriate markers (Fig. 1g, h). In CMTr1, but not CMTr2 mutants, cOMe levels on the first nucleotide are strongly reduced, but still detectable indicating that CMTr1 is the main methyltransferase and that only CMTr1/2 double knock-out flies completely lack cOMe at the first nucleotide (Fig. 1g, h).

To specifically analyze methylation of the first nucleotide in polyA mRNA, we decapped polyA mRNA with the pyrophosphatase RppH and removed the first phosphate for labeling the first nucleotide by $^{32}$P-gammaATP followed by digestion into individual nucleotides by nuclease P1 (Supplementary Fig. 2a) and separation on 2D thin-layer chromatography (TLC) (Fig. 1i). In S2 cells, we detected cOMe on adenosine (pAm) and cytosine (pCm, Fig. 1j, k), but in female flies predominantly pAm was present (Fig. 1k). By omitting decapping, residual rRNA in the polyA mRNA preparation was analyzed and this RNA does not show cOMe (Supplementary Fig. 2b). Gm runs at the same position as C, and thus can not be distinguished (Supplementary Fig. 2c) and Um runs at the same position as dT, which can be carried over as a contaminant of oligo dT purification (Fig. 1i)[9]. Although single mutants in CMTr1 or CMTr2 still had cOMe, the double mutants were devoid of cOMe further suggesting that these two enzymes have overlapping functions and are both able to methylate the 2′-O-ribose of the first transcribed nucleotide (Fig. 1l-n).

Our TLC analysis of the first nucleotide of mRNAs in Drosophila shows a strong preference for A from quantification of the four nucleotides in CMTr1/2 double knock-out flies (Fig. 1o). We validated the accuracy of the TLC data by analyzing CAGEseq from Drosophila. The CAGEseq data corroborated observations from the TLCs, demonstrating a strong preference for A as the first nucleotide in Drosophila mRNA (Fig. 1o) which is further consistent with the transcription initiator motif (Inr) sequence YYANWYY (Y: pyrimidine, N: any nucleotide and W: A or T) containing one A in the consensus sequence[30].

To further test that both Drosophila CMTrs can methylate the first nucleotide, we expressed Drosophila CMTr1 and CMTr2 in Drosophila S2 cells (Supplementary Fig. 2d). Both Drosophila CMTrs show equal activity in methylating the first nucleotide in vitro using a $^{32}$P-GTP capped RNA substrate with the consensus start sequence AGU after digestion with RNase I as judged by comparison to a single nucleotide ladder and appropriate markers (Supplementary Fig. 2e). Likewise, when the first nucleotide from this RNA (lanes 8 and 9) is labeled, only pAm is detected after digestion with nuclease P1 on 2D TLCs, which confirms that the first nucleotide of the substrate is A (Supplementary Fig. 2f).

***CMTrs are broadly expressed.*** Global expression studies of CMTr1 and CMTr2 showed that both are expressed throughout development in a broad range of tissues with elevated CMTr1 levels during early embryogenesis and a peak of both in pre-pupae (Supplementary Fig. 3a, b)[31,32]. Both CMTr1 and CMTr2 show higher expression in larval brains and to some extent in the adult nervous system and in ovaries (Supplementary Fig. 3b). CMTr2 is also highly expressed in the testis and trachea, which is consistent with a previously described transient role in tracheal development[33].

Analysis of expression from epitope-tagged genomic rescue constructs in the larval ventral nerve cord and adult brains revealed expression of both CMTr1 and 2 primarily in a pan-neural pattern with a predominantly nuclear localization compared to the nuclear neuronal marker ELAV (Supplementary Fig. 3c–n). To obtain a

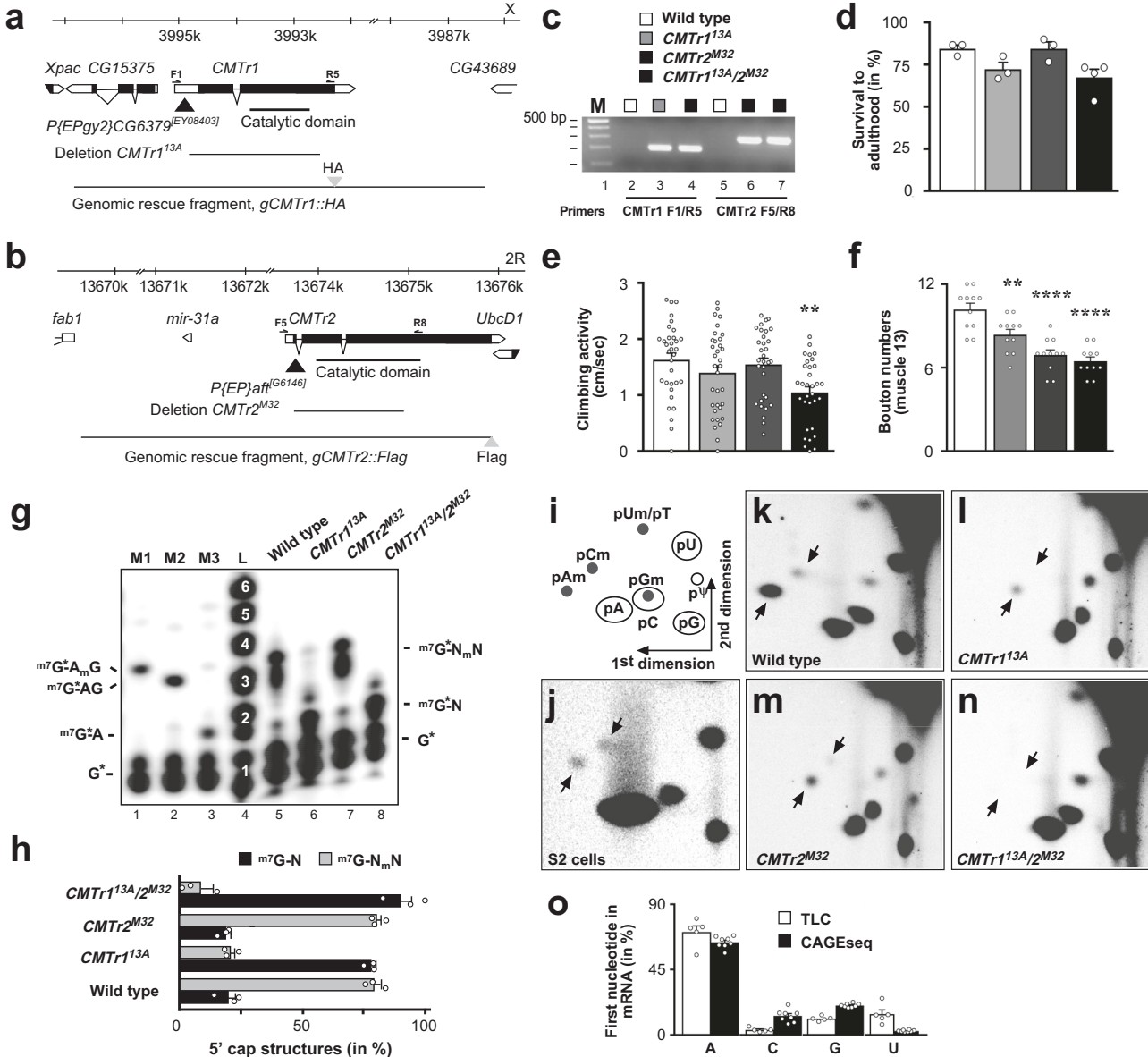

**Fig. 1 Analysis of CMTr1 and CMTr2 null mutants and mRNA cap 2′-O-ribose methylation in Drosophila. a**, **b** Genomic organization of the *CMTr1* and *CMTr2* loci depicting the transposons (black triangle) used to generate the deletions *13A* and *M32*, which are null alleles. Genomic rescue fragments tagged either with hemaglutinin (HA, **a**) or FLAG (**b**) epitopes are indicated at the bottom. Primers used for validating the deletions are indicated on top of the transcript. **c** Validation of *CMTr1[13A]* and *CMTr2[M32]* single and double mutants by genomic PCR. The gel is representative of two biological replicates. The marker is a 100 bp DNA ladder with 500 bp indicated on top. Wild type is indicated in white, *CMTr1[13A]* in light gray, *CMTr2[M32]* in dark gray, and *CMTr1[13A]/2[M32]* in black. **d** Survival of flies to adulthood after hatching from the eggshell shown as mean ±SE ($n = 3$, except *CMTr1[13A]/2[M32]* $n = 4$). **e** Climbing activity was assessed by negative geotaxis assays shown as mean ± SE, $n = 40$, $p = 0.005$ by one-way ANOVA followed by Tukey's test. **f** Bouton numbers at NMJs of muscle 13 in third instar larvae are shown as mean ± SE. $n = 11$, **$p = 0.005$ and ****$p \leq 0.0001$ by one-way ANOVA followed by Tukey's test. **g** Recapping of mRNA with $^{32}$PalphaGTP from adult flies of the indicated genotypes. 5′cap structures were separated on 20% denaturing polyacrylamide gels after digestion with RNAse I (lanes 5–8, right) Markers—M1: RNAse I digested $^{32}$PalphaGTP capped in vitro transcript starting with AGU and 2′-O-ribose methylated with vaccinia CMTr. M2: RNAse T1 digested $^{32}$PalphaGTP capped in vitro transcript starting with AGU. M3: RNAse I digested $^{32}$PalphaGTP capped in vitro transcript starting with AGU. Sequences of markers are shown on the left and of cap structures from adult flies are shown on the right, N: any nucleotide, *: $^{32}$P, m: methyl-group. L: single nucleotide ladder with nucleotide number indicated in white made by alkaline hydrolysis of a 5′ $^{32}$P-labeled RNA oligonucleotide. **h** Quantification of 5′ cap structures shown as mean ± SE. $n = 3$. Non-ribosemethylated cap is in black and ribose methylated in gray. **i** Schematic diagram of a 2D thin-layer chromatography (TLC) depicting standard and 2′-O-ribose methylated (m) phospho-nucleotides. ψ: pseudouridine. **j**–**n** Representative TLCs from three replicates showing modifications of the first cap-adjacent nucleotides of S2 cells (je), adult control (**k**), and *CMTr1[13A]* and *CMTr2[M32]* single (**l**, **m**) and double (**n**) mutant females. **o** Quantification of the mRNA first nucleotide shown as mean ± SE from TLC ($n = 5$, white) and CAGEseq data ($n = 8$, black) from adult *Drosophila* and S2 cells, respectively. Source data for gels, survival to adulthood, climbing activity, bouton numbers at muscle 13, for 5′cap structures, and first nucleotide in mRNA are provided as a Source Data file.

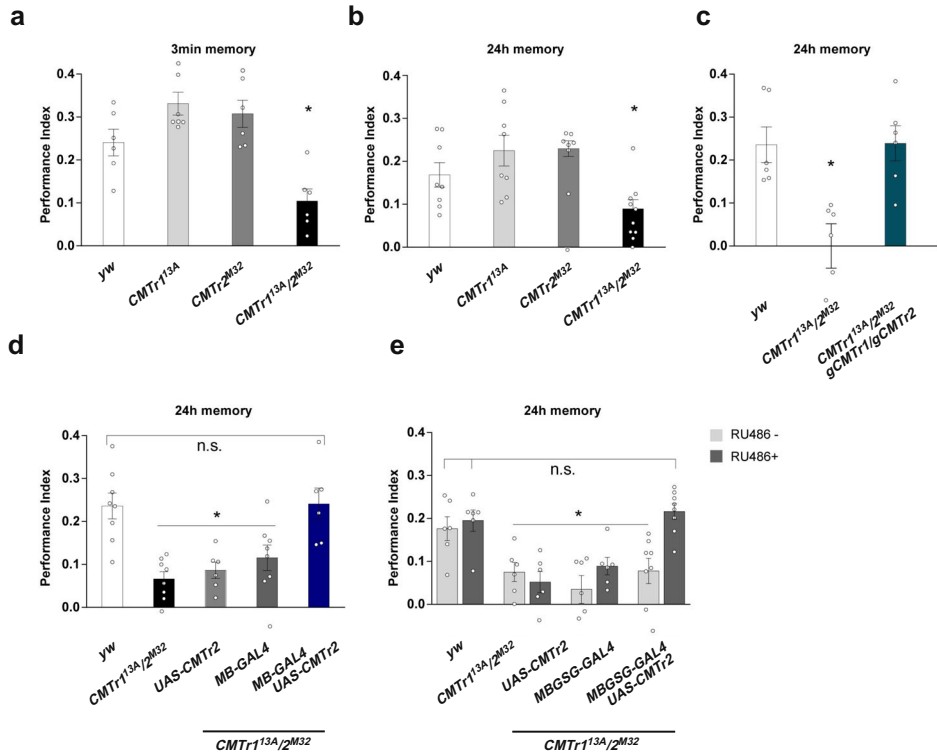

**Fig. 2 mRNA cap 2′-O-ribose methylation is required for reward learning in *Drosophila*. a, b** Appetitive memory immediately (**a**) and 24 h (**b**) after training of control (white) and *CMTr1*[13A] (light gray) and *CMTr*[M32] (dark gray) single and double mutant (black) flies shown as mean ± SE. $n = 8$ for A and $n = 6$ for B, except *CMTr1*[13A]/2[M32] $n = 10$, $p \leq 0.006$. **c** Rescue of the learning defect in *CMTr1*[13A]; *CMTr2*[M32] double mutant flies by genomic fragments (blue) shown as mean ± SE. $n = 6$, $p = 0.002$. **d, e** Rescue of the learning defect in *CMTr1*[13A]; *CMTr2*[M32] double mutant (black) flies by constitutive (**d**, $n = 8$, except *UAS-CMTr2* in light gray and *MB-GAL4 UASCMTr2* in blue in *CMTr1*[13A]/2[M32] $n = 6$, dark gray: *MB-GAL4*) or conditional (**e**, $n = 6$, except *MBGSG-GAL4 UASCMTr2* in *CMTr1*[13A]/2[M32] $n = 8$) expression of CMTr2 in mushroom bodies from *UAS* in the absence (light gray) or presence (dark gray) of RU486 shown as mean ± SE, $p \leq 0.0001$. Statistical analysis was done by one-way ANOVA followed by Dunnett's multiple comparison test. Source data for learning and memory experiments are provided as a Source Data file.

clearer view of the intracellular localization we stained epitope-tagged CMTr1 and CMTr2 in third instar salivary glands (Supplementary Fig. 3o–w). CMTr1, and to a lesser extent CMTr2, were both enriched in the nucleus but excluded from the nucleolus. There was also prominent localization of CMTr2 to the cytoplasm and the cell membrane, and this is also somewhat evident for CMTr1.

**Reward learning requires cap methylation**. mRNA modifications have been associated with neurological disorders and intellectual disabilities in humans[4,34]. Given the increased expression of CMTrs in the brain and their role in synapse differentiation (Fig. 1f, Supplementary Fig. 3a, b), we adopted a learning and memory paradigm as a sensitive assay to evaluate the molecular functions of cOMe in neurons. In particular, we used appetitive conditioning learning and memory assay whereby a sugar reward is paired with a specific odor because it rapidly induces protein-synthesis-dependent memory[35].

Immediate (3 min) and 24 h memory of single *CMTr1*[13A] and *CMTr2*[M32] mutant flies was indistinguishable from that of wild-type controls. However, both immediate and 24 h memory were significantly impaired in *CMTr1*[13A]; *CMTr2*[M32] double mutant flies (Fig. 2a, b). These memory performance deficits were restored by introducing transgenes encoding genomic fragments for both *CMTr1* and *CMTr2* (Fig. 2c), indicating that the learning deficits arise from the absence of CMTr function.

We also tested the performance of *CMTr1*[13A]; *CMTr2*[M32] double mutant flies using aversive olfactory conditioning which pairs one of two odors with an electric shock. Surprisingly,

aversive learning of *CMTr1*[13A]; *CMTr2*[M32] double mutant flies was indistinguishable from that of control flies, which suggests specificity for the reward learning defect (Supplementary Fig. 2a). We confirmed that mutant flies behave normally when exposed to the repellent odors and they can detect sugar (Supplementary Fig. 2b, c). These sensory controls and the wild-type aversive learning performance of *CMTr1*[13A]; *CMTr2*[M32] also suggest that CMTr deficiency somehow specifically impairs reward learning.

Olfactory learning and memory in *Drosophila* are coded within the neuronal network of the mushroom bodies (MBs)[36]. Valence learning can be coded as changes in the efficacy of synaptic junctions between odor-activated Kenyon Cells (KCs, the intrinsic cells of the MB) and specific mushroom body output neurons. We, therefore, tested whether the reward learning defect of *CMTr1*[13A]; *CMTr2*[M32] mutant flies could be rescued by restoring CMTr expression to KCs. Expressing a *UAS-CMTr2* transgene in the KCs using *MB247-GAL4* rescued the learning deficits of *CMTr*[13A]; *CMTr*[M32] double mutant flies (Fig. 2d).

Next, we investigated whether the reward learning phenotype of *CMTr*[13A]; *CMTr*[M32] double mutant flies arose from a developmental origin, or from loss of an acute function in the adult stage. The gross morphology of the adult MBs appears to be normal in *CMTr1*[13A]; *CMTr2*[M32] double mutants as judged from expressing a *UAS-EGFP* transgene with *MB247-GAL4*, or with the KC-subtype restricted drivers *NP7175-GAL4* (αβ core KCs), *0770-GAL4* (αβ surface KCs) or *1471-GAL4* (γ KCs, Supplementary Fig. 3a). Interestingly, restoration of *CMTr2* expression to these more restricted KC subsets did not rescue the learning defect of *CMTr*[13A]; *CMTr*[M32] double mutant flies (Supplementary Fig. 3b).

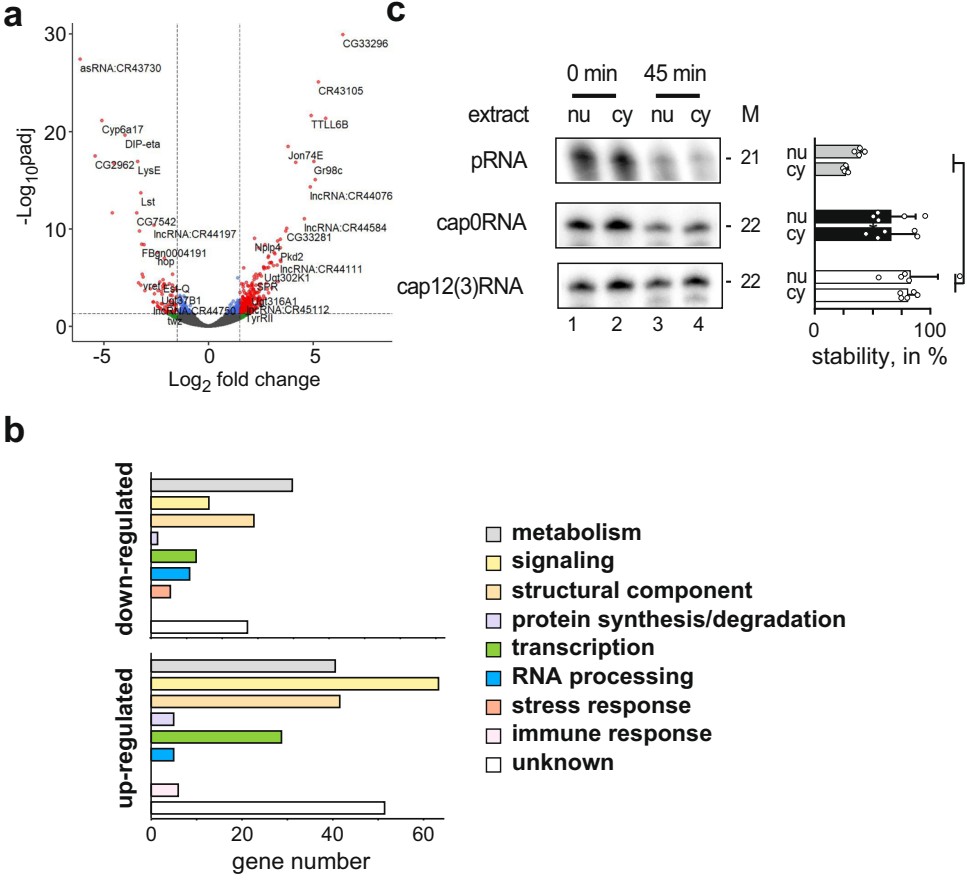

**Fig. 3 Impact of mRNA cap-adjacent 2′-O-ribose methylation on gene expression and RNA stability. a** Volcano plot depicting differentially expressed genes in *CMTr1^{13A}*; *CMTr2^{M32}* double mutant flies compared to control flies. **b** Functional classification of upregulated (bottom) and downregulated (top) genes in *CMTr1^{13A}*; *CMTr2^{M32}* double mutant flies compared to control flies. **c** Incubation of monophosphorylated RNA (pRNA, gray) and capped RNA with (white) or without 2′-O-ribose methylation of cap-adjacent nucleotides (black) in nuclear (nu) and cytoplasmic (cy) extracts from S2 cells. M: length in nucleotides. The graph to the right depicts the percent undegraded RNA left after 45 min as mean ± SE (*n* = 5, except *n* = 4 for pRNA, *p* = 0.03 by one way ANOVA followed by Tukey's test). Source data for gel and RNA stability values are provided as a Source Data file.

We next tested whether the reward learning defect of *CMTr^{13A}*; *CMTr^{M32}* double mutant flies could be rescued by inducing CMTr2 expression just before training in adult flies. Since *MB247-GAL4* was able to restore learning, we employed an *MB247*-driven Gene-Switch (GS) to conditionally induce CMTr2 expression by feeding flies with RU486. Only *CMTr^{13A}*; *CMTr^{M32}* double mutant flies that harbored the *MB247-GS* and *UAS-CMTr2* transgenes exhibited restoration of memory performance when fed with RU486 (Fig. 2e). Together these experiments suggest that CMTr in the MB KCs plays a key role in olfactory reward learning.

**CMTr loss increases the abundance of certain transcripts.** To investigate the impact of cOME on gene expression, we performed RNA sequencing on cOMe deficient and control flies. Differential gene expression analysis revealed 197 and 701 genes that were significantly downregulated and upregulated in *CMTr^{13A}*; *CMTr^{M32}* double mutant flies as compared to wild-type controls (adjusted *p*-value < 0.05, at least twofold change, Fig. 3a, Supplementary Data 1). GO term analysis revealed significant upregulation of genes involved in metabolism, receptor signaling, and cell adhesion (Supplementary Data 2). To obtain a high confidence list of significantly differentially regulated genes, we took genes threefold differentially regulated (80 and 244 genes downregulated and upregulated in double-

mutant flies compared to controls) and analyzed them according to gene function by annotated protein domains. This analysis confirms prominent effects on gene networks involved in metabolism, cellular signaling, and structural cell components, including a number of cell adhesion molecules. The complement of genes differentially expressed in *CMTr^{13A}*; *CMTr^{M32}* double mutant flies is qualitatively different from the loss of the other prominent mRNA modification m6A or from loss of the transcription factor *erect wing* that regulates synapse numbers (Fig. 3b, Supplementary Data 2)[37–39].

Notably, immune genes were not significantly upregulated in the double mutant flies (Supplementary Data 1) and CMTr1 knock-out mice[40]. In addition, cOMe is only about 30% in mice measured by TLC[9,40] and about 80% in *Drosophila* measured by recapping (Fig. 1g, h). For a primary role of cOMe for self/non-self discrimination, one would expect 100% cOMe. The relevance of cOMe to prevent detection of non-self RNA by the evolutionary younger vertebrate immune system is linked to the interferon response, which is absent in flies, and they also do not possess unmethylated cap RNA sensors Rig-I and IFITs[15].

A potential role of cOMe could be to stabilize mRNA transcripts. However, we find a 3.5-fold increase in up-regulated transcripts compared to downregulated transcripts in the absence of cOMe, which does not support a general role of cOMe in protecting mRNAs from degradation in *Drosophila*. To further test, whether cOMe protects mRNAs from degradation, we generated fully

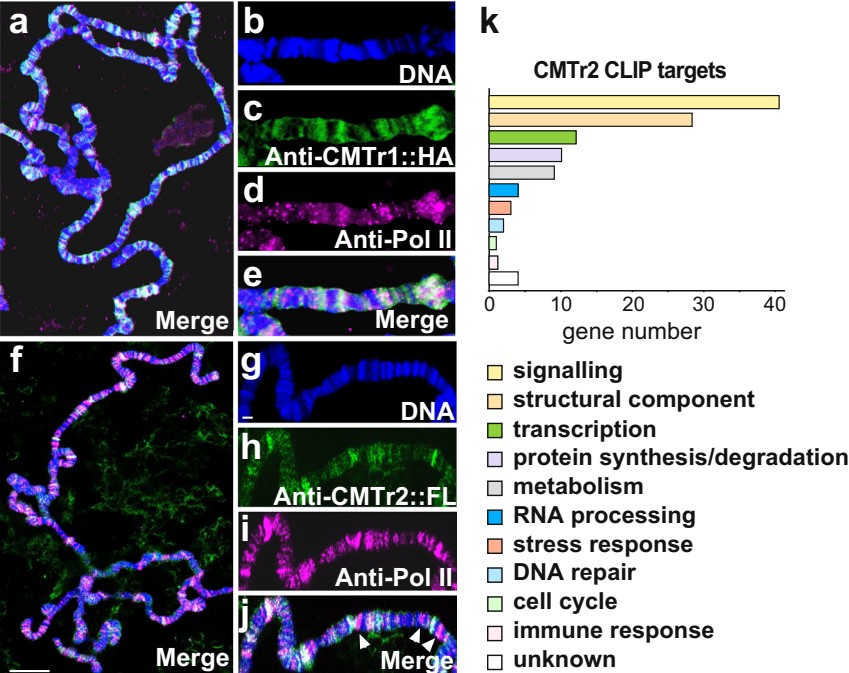

**Fig. 4 CMTr2 localizes to distinct sites of transcription and has a dedicated set of targets. a–j** Representative images of polytene chromosomes from salivary glands from three replicates expressing CMTr1::HA (**a–e**) or CMTr2::FLAG (**f–j**) stained with anti-Pol II (magenta, **d**, **i**), anti-HA (green, **c**, **h**), and DNA (DAPI, blue, **b**, **g**), or merged (white, **a**, **e**, **f**, **j**). Arrowheads indicate the absence of CMTr2. Scale bars in **f** are 10 μm and in **g** are 2 μm. **k** Functional classification of CMTr2 CLIP targets.

capped RNA oligonucleotide of the trypanosomal splice leader known to have cOMe with or without methylation using the vaccinia capping enzymes and noted that vaccinia CMTr can 2′-O-methylate the ribose of the first three nucleotides (Supplementary Fig. 6). When we incubated these RNA oligonucleotides that were uncapped, capped, and capped with cOMe in nuclear and cytoplasmic *Drosophila* S2 cell extracts, cOMe did not affect RNA stability, but whether this is due to methylation of the first three nucleotides and is sequence-specific needs to be determined in follow up studies. Lack of a cap resulted in increased degradation, which is consistent with observations in mammalian systems[41] (Fig. 3c).

**CMTr2 has a dedicated set of target genes**. We next investigated how many genes produce mRNAs that contain cOMe. We reasoned that cOMe is either co-transcriptionally added to mRNAs of only a few specific genes or, of only a fraction of all mRNAs. To distinguish between these two possibilities, we stained polytene chromosomes from larval salivary glands.

CMTr1 prominently co-localized with RNA Pol II (Fig. 4a–e), suggesting that cOMe is introduced co-transcriptionally and is widespread. In contrast, CMTr2 is only prominently localized to a subset of transcribed genes suggesting that CMTr2 has a preferred set of target genes (Fig. 4f–j) and its association with polytene chromosomes does not expand in the absence of CMTr1 (Supplementary Fig. 7).

We subsequently used CLIP (cross-linking and immunoprecipitation) to identify targets for CMTr1 and CMTr2. For these experiments, we used a *CMTr* double knock-out line which contained genomic rescue constructs for *CMTr1* and *CMTr2* that are tagged with an HA or FLAG epitope, respectively. From these experiments we obtained 3109 and 762 genes for CMTr1 and CMTr2, respectively, that were twofold or more enriched above input (Supplementary Data 3 and 4). The larger number of CLIP targets for CMTr1 is consistent with broader staining on polytene

chromosomes, while CMTr2 introduces cOMe to a more specific set of target transcripts.

To obtain a high confidence catalog of CMTr1 and CMTr2 CLIP targets, we took genes that were at least 3-fold enriched (1146 and 117 genes, respectively) and analyzed them according to gene function. Consistent with previous analysis of differentially expressed genes (Fig. 3a, b, and Supplementary Data 1 and 2), this analysis revealed prominent effects on gene networks involved in cellular signaling including a number of genes encoding ion channels or their regulators in both CMTr1 and CMTr2, and also for genes involved in synaptic vesicle release and cell adhesion in CMTr2 (Fig. 4k, Supplementary Data 3 and 4).

**Cap methylation enhances the transport of untranslated mRNAs to synapses**. In the nucleus, the cap is bound by the CBC consisting of CBP20 and CBP80, which is replaced for the rate-limiting translation initiation factor eIF4E upon export to the cytoplasm and then followed by a pioneer round of translation[27,29]. Since cOMe only minimally enhances binding of eIF4E to the mRNA cap[42], we tested whether cOMe affects binding of the CBC complex to the mRNA cap by immunoprecipitation of CBP80 from nuclear extracts. Intriguingly, cOMe significantly increased the binding of CBP80 to capped RNA (Fig. 5a).

Since CBP80 prominently localizes to synapses in rats[43], we hypothesized that cOMe might mark certain mRNAs to maintain translational repression during transport to synapses for local translation[44,45]. To test whether we can detect the CBC at synapses in *Drosophila*, we pre-synaptically expressed epitope-tagged CBP20 from a *UAS* promoter with the neuronal *elav*[C155]-*GAL4* driver. In addition, we stained CBP80 with an antibody. Both CBP20 and CBP80 were found to be localized to synapses at the NMJs of third instar larvae, suggesting the presence of untranslated mRNAs. In *CMTr* double mutant larvae, however, localization of both CBP20 and CBP80 is reduced at the site of transmitter release (active zones) marked by the presence of

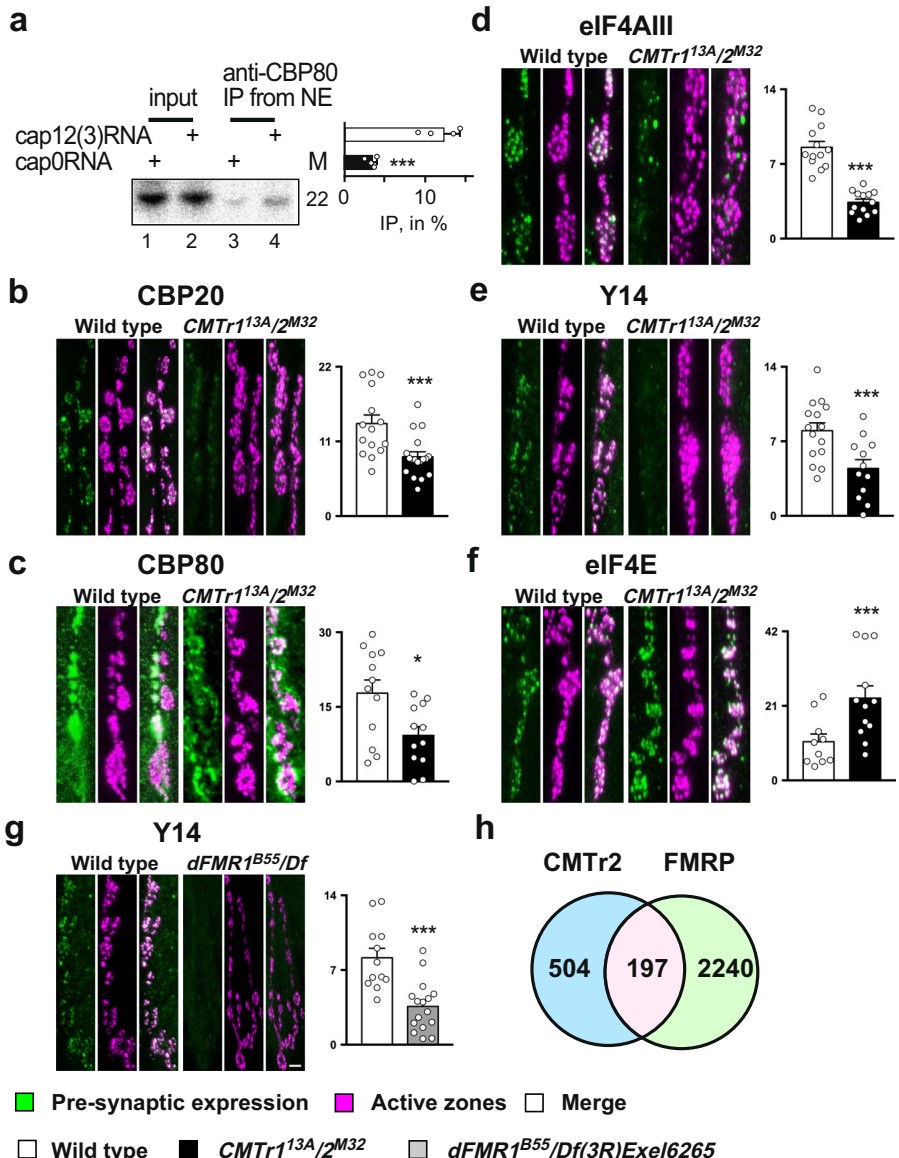

**Fig. 5 2′-O-ribose methylation of mRNA cap-adjacent nucleotides is required for localizing untranslated transcripts to synapses. a** Immunoprecipitation of capped RNA with or without 2′-O-ribose methylation from nuclear extracts. M: length in nucleotides. The graph to the right depicts the ratio of IP/input ($n = 4$, $p = 0.016$). **b–g** Representative images from staining of synapses at third instar NMJs pre-synaptically expressing epitope-tagged markers with *elav*[C155]-*GAL4* from *UAS* or stained with an antibody (CBP80) in control (left, green) and *CMTr1*[13A]; *CMTr2*[M32] double mutant larvae (**b-f**) or *dFMR1*[B55]/*Df(3R)Exel6265* mutant larvae (**g**, right, green). The active zone of synapses was stained with nc82 (magenta). The mean ± SE of the intensity is shown on the right is arbitrary units in white for the control, in black for *CMTr1*[13A]; *CMTr2*[M32] double mutant larvae (**b-f**) and in gray *dFMR1*[B55]/*Df(3R)Exel6265* mutant larvae (**g**) for nuclear cap-binding proteins CPB20 (**b**, $n = 16$ and 19, ***$p = 0.0015$) and CBP80 (**c**, $n = 12$, *$p = 0.05$), for nuclear exon junction complex (EJC) proteins eIF4AIII (**d**, $n = 13$, ***$p = 0.005$) and Y14 (**e**, $n = 15$ and 12, ***$p = 0.003$), for the rate-limiting translation initiation factor eIF4E (**f**, $n = 10$ and 12, ***$p = 0.009$) and for nuclear exon junction complex (EJC) protein Y14 in control and *dFMR1*[B55]/*Df(3R)Exel6265* mutant larvae (**g**, $n = 12$ and 16, ***$p = 0.005$). The scale bar in **g** is 1 μm. **h** Overlap (pink) of CMTr2 CLIP targets (blue) with FMRP targets (green) in *Drosophila*. Statistical analysis was done by an unpaired *t*-test. Source data for gel, immunoprecipitations values, and intensity of pre-synaptic bouton stainings are provided as a Source Data file.

Bruchpilot (nc82) (Fig. 5b, c). We also note that CBP80 is present post-synaptically in *Drosophila*, although at the periphery of the synaptic bouton (Fig. 5c). In the soma, CBP20 localizes to both the nucleus and cytoplasm (Supplementary Fig. 8a).

To validate whether CMTrs are required for localization of translationally repressed mRNAs to synapses, we stained NMJs for pre-synaptically expressed exon junction complex (EJC) proteins Y14 and eIF4AIII. EJC proteins are predominantly nuclear and deposited during splicing to ensure mRNA quality control by nonsense-mediated decay[46]. Upon export from the

nucleus, the EJC is stripped off from the mRNA upon the first round of translation[29].

Consistent with a pool of untranslated mRNAs at synapses, we find both Y14 and eIF4AIII proteins localized at synapses (Fig. 5d, e). In the absence of CMTrs, both Y14 and eIF4AIII are also reduced at synapses similar to CBC20 and CBC80. We propose that CMTrs are required for translational silencing during the transport to synapses until they are translated. In contrast, the rate-limiting translation initiation factor eIF4E is present at synapses, and levels are increased, possibly as compensation for reduced translation due to less mRNA

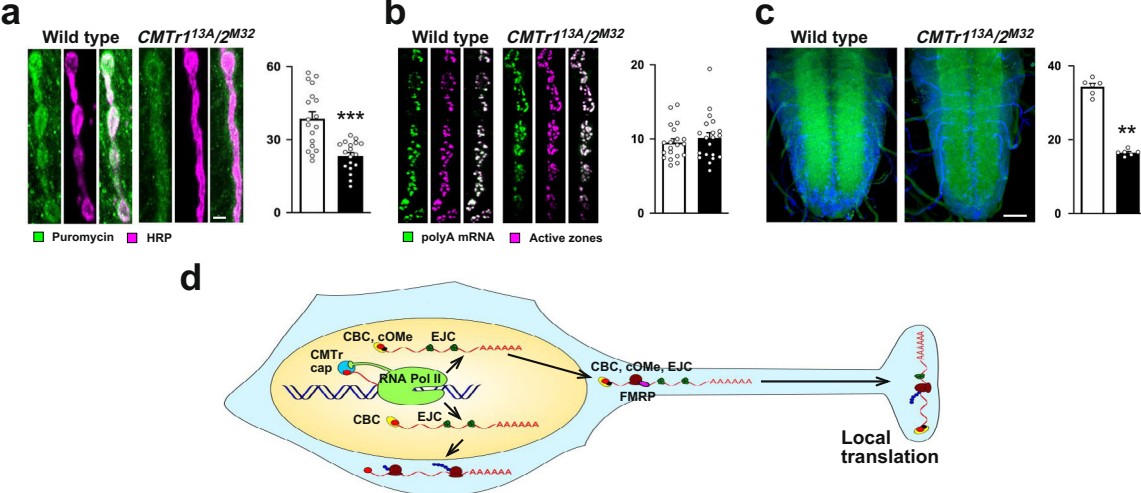

**Fig. 6 2′-O-ribose methylation of mRNA cap-adjacent nucleotides enhances translation at synapses. a** Staining of synapses at third instar NMJs with anti-puromycin antibodies (left, green) and with anti-HRP recognizing a neuronal epitope (right, red) after 30 min of puromycin incorporation in control and *CMTr1^13A*; *CMTr2^M32* double mutant larvae. The mean ± SE of the intensity is shown on the right is arbitrary units in white for the control, in black for *CMTr1^13A*; *CMTr2^M32* double mutant larvae (n = 18, ***p ≤ 0.001). **b** RNA in situ hybridization with an oligo(dT) probe to mRNA in synapses at third instar NMJs (left, green) in control and *CMTr1^13A*; *CMTr2^M32* double mutant larvae. The active zone of synapses was stained with nc82 (magenta, right). The mean ± SE of the intensity is shown on the right is arbitrary units in white for the control, in black for *CMTr1^13A*; *CMTr2^M32* double mutant larvae (n = 20). **c** RNA in situ hybridization with an oligo(dT) probe (green) to mRNA in the ventral nerve cord of third instar larvae (green) in control and *CMTr1^13A*; *CMTr2^M32* double mutant larvae counterstained with DAPI (blue). The mean ± SE of the intensity is shown on the right is arbitrary units in white for the control, in black for *CMTr1^13A*; *CMTr2^M32* double mutant larvae (n = 6, **p = 0.002). Scale bars in **a** and **c** are 1 μm and 50 μm, respectively. Statistical analysis was done by an unpaired *t*-test. **d** Model for the role of cap-adjacent 2′-O-ribose methylation in gene expression in neurons. EJC exon junction complex containing Y14 and eIF4AIII, CBC cap-binding complex consisting of CBP20 and 80, CMTr cap methyltransferase, FMRP Fragile X Mental Retardation Protein, cOMe cap 2′-O-ribose methylation at cap-adjacent nucleotides, ribosomes are shown as brown blobs. Source data for the intensity of pre-synaptic bouton and brain stainings are provided as a Source Data file.

at synapses (Fig. 5f). In the soma, eIF4E predominantly localizes to the cytoplasm (Supplementary Fig. 8b).

Fragile X Mental Retardation Protein (FMRP, dFRP1 in *Drosophila*) is highly expressed in the brain with important functions in synaptic plasticity and neurodevelopment. FMRP has been shown to repress the translation of specific mRNAs by stalling ribosomes[47–49].

To validate that lack of translational repression reduces localization of untranslated mRNAs to synapses, we analyzed the localization of Y14 to synapses in dFMR1 mutants. Indeed, levels of the Y14 marker for untranslated mRNAs are dramatically reduced at synapses of dFMR1 mutants due to premature translation (Fig. 5g).

Since FMRP repression of translation could be connected to cOMe, we analyzed the overlap in targets between FMRP determined for cholinergic and GABAergic neurons in *Drosophila*[50] and CMTr2 CLIP targets. Among the CMTr2 CLIP targets, 28% have also been identified as FMRP targets in this subset of neurons (p ≤ 0.00001, Fig. 5h). In contrast, only 2% of CMTr2 targets are identified as FMRP targets in non-neuronal *Drosophila* S2 cells (Supplementary Data 5).

Since levels of untranslated mRNAs are reduced at synapses in the absence of cOMe, we anticipate that translation at synapses is also reduced. To validate that lack of CMTrs leads to reduced translation at synapses, we measured puromycin incorporation by antibody staining with anti-puromycin antibody and compared puromycin incorporation levels to the neuronal epitope recognized by anti-HRP antibody. Consistent with the reduced amount of untranslated mRNAs at synapses at third instar NMJs, we also find that nascent translation is significantly reduced at synapses in *CMTr1^13A*; *CMTr2^M32* double mutant flies (Fig. 6a), but not in the soma of ventral nerve cord neurons (Supplementary Fig. 8c). For all mRNA detected by RNA in situ with an oligo(dT) probe,

we did not find a reduction at synapses at NMJs of third instar larvae in *CMTr1^13A*; *CMTr2^M32* double mutant flies (Fig. 6b), but mRNA levels in the neuropil of the larval ventral nerve cord were significantly reduced in *CMTr1^13A*; *CMTr2^M32* double mutant flies (Fig. 6c).

## Discussion

Although known for over 40 years, the role of cOMe in animals has been enigmatic due to the lack of knockout models[15]. Here, we show that CMTr1 is the major enzyme responsible for introducing cOMe in *Drosophila*, which has also been found in knock-out mice[40,51], but only in double-knockout *Drosophila* cOMe is absent. It has previously been postulated that CMTr1 methylates the ribose of the first, and CMTr2 the second nucleotide[6,20,21]. For both *Drosophila* CMTrs, however, we find that each can methylate the first nucleotide of mRNAs in vivo and in vitro. Since vaccinia CMTr methylates the first nucleotide in transcripts with a consensus AGU start but methylates the first three nucleotides in a trypanosomal splice leader starting with AACU, it is conceivable that CMTrs methylate differently depending on sequence context which could explain differences between studies[20]. In addition, we have developed a battery of standards to determine the extent of methylation of the first nucleotides in mRNA. Since CMTr2 co-transcriptionally associates only with a subset of genes visualized on polytene chromosomes and seems to primarily target transcripts starting with A (this study and[51]), this further indicates target specificity, but new sensitive assays need to be developed for a complete analysis[5].

Loss of CMTrs has few obvious phenotypic consequences leading to the development of healthy and fertile flies, while CMTr1 in mice is essential and required for neurogenesis[40]. In accordance with the prominent expression of mRNA

methyltransferases in the brain[3,4], however, we find that CMTrs are essential for reward learning and that CMTrs are redundant in this process.

## Tuning protein synthesis in neurons for learning.

Short-term reward memory measured immediately after training is considered to be insensitive to blockers of protein synthesis[35,52]. It, therefore, seems somewhat enigmatic that CMTrs would play an acute role in the reward learning process. Moreover, cOMe occurs in the nucleus before the mRNAs undergo a lengthy journey to the synapse. Our experiments demonstrate a role for cOMe in adult KCs but the two days required to induce CMTr2 expression do not have the required temporal resolution to distinguish between roles before and during learning itself. We therefore currently favor a model for CMTrs in establishing/maintaining the appropriate repertoire of locally-translated synaptic factors in adult KC synapses, that are necessary to support reward learning, rather than directly in learning-induced synaptic change. Consistent with prior reports of neuronal localization of mRNAs encoding cytoskeletal proteins, neurotrophins, membrane receptors, and regulatory kinases important for synaptic activity and plasticity[53], we find that CMTr targets include many cell adhesion and signaling molecules. Of note, the *volado*-encoded α-integrin that was shown to be defective in short-term memory performance is a CMTr2 target[54]. Work in several organisms has also demonstrated roles for neuronal cell adhesion molecules (NCAMs) in acute forms of plasticity and includes *Drosophila* mutants in the N-CAM homolog fasII[55,56]. Although both of these *Drosophila* studies revealed defects in short-term aversive memory, other locally translated adhesion molecules could also be specifically required to support short-term and more persistent reward memory.

## Novel routes for local translation.

It is well-known that many mRNAs are transported and stored in various cellular locations including dendrites and synapses[45,53]. In dendrites, translation of mRNAs occurs in polysomes, while in synapses the main form of translation is from monosomes[57]. A feasible way to transport ribosomes to synapses is to arrest them on mRNAs and only allow translation once they have arrived at synapses. Indeed, FMRP has been shown to arrest ribosomes[47,48]. Accordingly, the absence of FMRP would lead to premature translation. Our data support this model as the EJC protein Y14, a marker for untranslated mRNAs, is drastically reduced at synapses in FMRP mutants. Likewise, we find a substantial overlap of CMTr2 CLIP targets with FMRP targets. Further support of such a model also comes from our finding that the level of translation initiation factor eIF4E is not reduced at synapses demonstrating that this is not the consequence of generally reduced transport. Also, eIF4E is transported separately from mRNAs as it localizes with P body markers in neuronal processes, while CBC and EJC proteins colocalize in distinct granules[58].

Our observation, that CBC and EJC proteins are not completely missing from synapses in the absence of cOMe is consistent with the presence of multiple pathways for regulating local translation at synapses[59]. The key to distinct pathways might rely on information encoded in different promoters that are incorporated co-transcriptionally through the presence of CMTrs at sites of transcription.

In essence, our discovery of a function for CMTrs in learning and localization of untranslated transcripts to synapses for local translation (Fig. 6d) has important implications in understanding the role of these modifications in affecting gene expression in synaptic plasticity.

## Methods

**Generation of mutant fly strains.** The deletion allele *y w CMTr1[13A]* (excision 13A) and *y w; CMTr2[M32]* (excision M32) were obtained from imprecise excision of transposon *P{EPgy2}CG6379[EY08403]* over *Df(X)BSC869* and *P{EP}aft[G6146]* over *Df(2 R)BSC347* in females and mapped by primers CG6379 F1 (GTCTGGACTT ATCGCACCACCTATCG) and R5 Spe (GGTAACTAGTGCTGTGGCCCAAC TTGTCCGCAATGAAC), and aft F5 (CCTTCCGAAGTGGAGCAGCTCTTCG AG) and R8 (GGTGGCAGGTAGCATAGTGTCTTGCTTTC). The 192 bp and 287 bp PCR fragments were sequenced for validation. *y w CMTr1[13A]* and *y w; CMTr2[M32]* excision lines were viable when first generated. To normalize genetic backgrounds, excision lines were outcrossed to the Df lines for five generations. A control *y w* line was generated by crossing *Df(X)BSC869* to *P{EP}aft[G6146]* and *Df(2R)BSC347* to *P{EPgy2}CG6379[EY08403]* for five generations and then combined. To determine survival of mutants, freshly hatched larvae were individually picked and grown in groups of 30 and surviving adults counted.

**Generation of constructs and transgenic fly strains.** To clone *CMTr1* and *CMTr2* cDNAs, total RNA was extracted with Tri-reagent (SIGMA) from larval brains and reverse transcribed with Superscript II as described[60]. *CMTr1* was amplified from this cDNA with primers pUAST CG6379HA F2 (CGAACCT TCGGACGATGAGAACTCGGAGCCCACGCCCAAGAAG) and pUAST C G6379 F3 (GCAGAATTCGAGATCTAAAGAGCCTGCTAAAGCAAAAAG AAGTCACCATGGACGAACCTTCGGACGATGAGAACTCG) with return primer R5 Spe in a nested PCR with Q5 polymerase (NEB) and cloned with EcoRI and SpeI into a modified pUAST vector containing an attB site for phiC31 mediated integration. The w+-marked *pUAST CMTr1:HA* construct was inserted into attP VK0002 at 76A by phiC31 transgenesis.

*CMTr2* was amplified from this cDNA as two fragments with primers aft cDNA F1 (CCTGCTAAAGCAAAAAAGAAGTCACCATGAGCTTTCGTTCGTCTCC GCAGGGAAAGCCAC) and aft cDNA F2 (GGGAATTCGAGATCTAAAGAGC CTGCTAAAGCAAAAAAGAAGTCACCATG) as nested PCR and aft cDNAR2 (CTCATCCTTTTCATATTTGCTATGAAGGTAATGATTCAGAGATGCTATG), and the second fragment with aft cDNA F3 (TACCTTCATAGCAAATATGAA AAGGATGAGATTAAATGGCGCTGGCGCTCAACTACTTTG) and aft cDNA R1 (CTCGGTACCAAATACtGCTGCCGACTCTTGGATGGAACCGACATCTG) with Q5 polymerase (NEB), the two PCR fragments were then fused by PCR and cloned with EcoRI and KpnI into the pUC 3GLA vector[61] containing an attB site for phiC31 mediated integration. The GFP+ -marked *pUC 3GLA UAS CMTr2:FLAG* construct was inserted into attP40 at 25C by phiC31 transgenesis.

Genomic rescue constructs were made by recombineering from BAC clones. For gCMTr1, the ends were amplified with Q5 polymerase (NEB) using primers dMtr end1F1 (GGCACTAGTgcgcatgaattaagtgctaaaatgtg) and dMtr end1R1 (ATCCCGGCTTATGTGTGTCCAACATG), and dMtr end2F2 (ATCCCAAAC CGAACCACATTAAAGG) and dMtr end2R2 (CCGTGGTACCGGTGTTATG CTCGGACAGTGGTAATCGAATG) from BAC DNA prepared as described[62] and cloned into pUC 3GLA using SpeI and KpnI. The 10.5 kb genomic fragment was then retrieved using the ends vector linearized with EcoRV from BacR21I10 as described[61]. The C-terminal HA TEV myc tag was then incorporated by PCR into a 495 bp AvrII and SbfI fragment and cloned with these sites. The GFP+-marked *pUC 3GLA gCMTr1:HATEVmyc* construct was inserted into attP VK0022 at 57F by phiC31 transgenesis.

For gCMTr2, the ends were amplified with Q5 polymerase (NEB) using primers aft end1 F1 Bam (CCAGGATCCGCGGCCGCATGGGAGGTATGCGATTAATG GC) and aft end1 R1 Xba (CCTCTAGAGGCCTAAATTTGAAATAGTTATCTCC ATATAATATTTATGAG), and aft end2 F2 Xba (GCCTCTAGAGGCCTGTTTCT CACCCATTACGC) and aft end2 R2 PvuII (CTGATCCCTGGAAGTAAAGATT CTCGGTACCAAATACTGCTGCCGACTCTTGGATGGAAC) from BAC DNA and cloned together with a linker BirA FLAG linkA (CTGGAGGATTAAATGA CATCTTTGAAGCACAGAAGATCGAATGGCATGAGGATTACAAGGACGA CGATGACAAGGCTTGA) and BirA FLAG linkB (CTAGTCAAGCCTTGTCATCGTCGTCCTTGTAATCCTCATGCCATTCG ATCTTCTGTGCTTCAAAGATGTCATTTAATCCTCCAG) into a modified pUAST using BamHI and SpeI in a four-way ligation. The 6.7 kb genomic fragment was then retrieved using the ends vector linearized with StuI from BacR20E20 as described[61]. The w+-marked *CASPR gCMTr2: TEVFLAG* construct was inserted into attP40 at 25C by phiC31 transgenesis.

For expression in S2 cells, *CMTr 1* and *CMTr 2* were subcloned into a *Drosophila* actin promoter containing plasmid *pAct* [39]by introducing N-terminal Strep and FLAG tags using primers dCMTr1 F1 (GATTACAAGGATGACGATGA CAAGGCCTCTGACGAACCTTCGGACGATGAGAACTC) dCMTr1 R1 (GTGGAGATCCATGGTGGCGGAGCTCGAGCTAGCTGTGGCCCAA CTTGTCCGCAATGAAC), and dCMTr2 F1 (GATTACAAGGATGACGATGA CAAGGCCTCCTTTCGTTCGTCTCCGCAGGGAAAGCCAC) and dCMTr2 R2 (GTGGAGATCCATGGTGGCGGAGCTCGAGTTAAAATACTGCTGCCGA CTCTTGGATGGAAC). For a pGEX fusion protein For expression in E. coli, dCMTr cDNA was cloned with primers GEX dCMTR12 F1 (CGACGTGCCCG ACTACGCAAGCCCCGGGCAAAAAGAAGTCACATGAGCGCTTGGTC) and GEX dCMTR1 R1 (TCGTCAGTCAGTCACGATGAATTGCGGCCGCTCTAG ACTAGCTGTGGCCCAACTTGTCCG) into a modified pGEX.

Essential parts of all DNA constructs were sequence verified.

**Behavioral assays**. For negative geotaxis experiments, groups of 20 flies kept in two inverted fly vials (19 cm) were tapped to the bottom. A movie was then made to record the moving flies and a frame about 5 s after the flies started running upwards and before the first fly reached the top was taken to measure the distance the flies have run upwards.

For learning and memory experiments, 2- to 5-day-old flies of both sexes were used for behavioral experiments in a T-maze. The odors used were 4-methylcyclohexanol (MCH) and 3-octanol (OCT).

For appetitive learning and 24 h memory testing, flies were starved for 21–23 h prior to training, and training was done as described[35]. Briefly, a group of about 120 flies were exposed first to the unconditioned odor (CS−) for 2 min followed by 30 s of air, and then to the conditioned odor (CS+) in the presence of dry sucrose for two minutes. For appetitive learning or immediate memory, flies were tested immediately after training for their choice between the two odors. For 24 h memory, flies were transferred into a standard cornmeal food vial after training and after one hour, they were transferred into food-deprivation vials until testing on the next day.

Odor and sugar acuity tests were performed as described in ref. [63] with some modifications. For odor acuity tests, starved flies were directly placed into the T-maze to test for odor avoidance (OCT or MCH) against the smell of plain mineral oil. For the sugar acuity test, a filter paper with size $18 \times 8$ cm was placed into a glass milk bottle (250 ml). Half of the filter paper ($\sim 9 \times 8$ cm) was soaked with saturated sucrose and dried before use. For the test, starved flies were placed into the bottle and the number of flies on both parts of the filter paper was counted separately 2 min later. The performance index was calculated as $[N_{sugar}/N_{total}] \times 100$, where $N_{total} = N_{sugar} + N_{plain}$.

For conditional expression GSG, GAL4 was used[64], which is activated by feeding flies with the progestin, mifepristone (RU486). Accordingly, flies were kept on RU486 (200 µM) (SIGMA), 5% ethanol) or control (5% ethanol) standard fly food for two days at 18 °C before starvation and training.

**Statistical analysis of behavioral data**. Behavioral data were analyzed using GraphPad Prism 7. Two-tailed $t$ tests were used for comparing two groups, and one-way ANOVA followed by a Tukey's post hoc test was used for comparing multiple groups.

**Analysis of cap-adjacent 2′-O-ribose methylation**. Total RNA was extracted with Trizol (Invitrogen) and polyA mRNA was prepared by oligo dT selection according to the manufacturer (Promega). For the analysis of 5′ cap structures, 150 ng of polyA mRNA by yDcpS in 30 µl for 1 h at 37 °C according to the manufacturer's instruction (NEB), then 54 µl AMPure XP magnetic beads (Beckman Coulter) were added and the decapped mRNA purified according to the manufacturer's instructions. The RNA was then eluted with 24 µl DEPC treated water and 3 µl capping buffer (NEB), 1.5 µl SAM (2 mM), 0.5 µl [α-32P] GTP (3000 Ci/mmol, 6.6 µM; Hartmann Analytic, Germany), 0.5 µl RNASe Protector (Roche) and 0.5 µl capping enzyme (NEB) was added and incubated for 1 h at 37 °C. The RNA was then purified as before and eluted in 10 µl DEPC treated water. An aliquot was then digested with RNASe I in NEB buffer 3 for 2 h and products were analyzed on 20% denaturing polyacrylamide gels pre-run for 2 h. Methylation levels were calculated as follows: $^{m7}G^* - N_mN/(^{m7}G^* - N + ^{m7}G^* - N_mN)$.

For the analysis of the first nucleotide in mRNA polyA mRNA from two rounds of oligo dT selection was used. Alternatively, polyA mRNA from one round of oligo dT selection was followed by ribosomal RNA depletion using biotinylated oligos as described[65] or by removal of rRNA by terminator nuclease (Epicenter) according to the manufacturer's instructions. For each sample, 50 ng of mRNA was decapped using either tobacco acid pyrophosphate (250 U; Epicenter) or RppH (NEB) in a buffer provided by the supplier and then dephosphorylated by Antarctic phosphatase (NEB). The 5′-end of dephosphorylated mRNAs were then labeled using 10 units of T4 PNK (NEB) and 0.5 µl [γ-32P] ATP (6000 Ci/mmol, 25 µM; Perkin-Elmer). The labeled RNA was precipitated, and resuspended in 10 µl of 50 mM sodium acetate buffer (pH 5.5) and digested with P1 nuclease (SIGMA) for 1 h at 37 °C. Two microliters of each sample were loaded on cellulose F TLC plates ($20 \times 20$ cm; Merck) and run in a solvent system of isobutyric acid:0.5 M NH$_4$OH (5:3, v/v), as first dimension, and isopropanol:HCl:water (70:15:15, v/v/v), as the second dimension. TLCs were repeated from biological replicates. The identity of the nucleotide spots was determined as described[9,66]. For the quantification of spot intensities on TLCs, a storage phosphor screen (K-Screen; Kodak) and Molecular Imager FX in combination with QuantityOne software (BioRad) were used.

For the analysis of CAGEseq data, nucleotides in the N1 position of mRNA following the m7G artifact were counted in a loop using grep in bash on all fastq files available from SRP131270 (data GSE109588)[30]. Counting lines with the pattern "…CAGCAGGN………." where N was replaced with the nucleotide being counted. Similarly, nucleotides were counted in the m7G artifact position using grep with the pattern "…CAGCAGN………." and in the second position following the m7G artifact using "…CAGCAGG.N………".

**Generation of S2 cell extracts**. S2 cells (ATCC) were cultured in Insect Express medium (Lonza) with 10% heat-inactivated FBS and 1% penicillin/streptomycin. Extracts were made after the Dignam protocol with modifications[67,68]. Cells were

washed in PBS and resuspended in five times the packed cell volume in buffer A (15 mM HEPES, pH 7.6, 10 mM KCl, 5 mM MgCl$_2$, 350 mM sucrose, 0.1 mM EDTA, 0.5 mM EGTA, 1 mM DTT, 1 mM PMSF (stock 0.2 M in isopropanol), 1 µg/ml leupeptin), spun down with 3000$g$ for 5 min and resuspended in buffer A and allowed to swell for 10 min on ice. Cells were then homogenized with a Dounce homogenizer with the loose pestle (B) with approximately 15 up and down strokes until cells were 80–90% lysed. The extract was then spun at 4000$g$ for 15 min, the supernatant was taken off and 0.11 volume buffer B (10×: 0.3 M HEPES, pH 7.6, 1.4 M KCl, 30 mM MgCl$_2$,) added. The supernatant was then spun at 34,000$g$ for 1 h. The resulting supernatant is the S-100 cytoplasmic extract. The nuclei were resuspended in 50% of the volume in buffer C (20 mM HEPES, pH 7.6, 420 mM NaCl, 1.5 mM MgCl$_2$, 0.2 mM EDTA, 0.5 mM DTT, 1 mM PMSF (stock: 0.2 M in isopropanol), 1 µg/ml leupeptin, 25 % v/v glycerol (Ultrapure, Gibco) using a pipette, a stirrer added, the volume slowly increased by another 50% of the nuclei volume with buffer C and then the nuclei were extracted for 30 min. The extract was then spun 30 min at 10,000$g$ at 4 °C and the supernatant was taken off without the white slur on top. This extract was then dialyzed in buffer E (20 mM HEPES, pH 7.6, 100 mM KCl, 0.2 mM EDTA, 0.5 mM DTT, 1 mM PMSF (stock: 0.2 M in isopropanol), 1 µg/ml leupeptin, 20 % v/v glycerol (Ultrapure, Gibco) for 2 hours. After dialysis, the supernatant was spun at 10,000$g$ for 10 min, and aliquots were frozen in liquid nitrogen, and extracts were stored at −80 °C.

**Generation of a cap labeled probe, RNA stability assay, immunoprecipitation, and denaturing gel electrophoresis**. As a probe for UV-crosslinking, RNA stability, and binding experiments the trypanosome spliced leader oligo (trypSL, AACUAACGCUAUUAUUAGAAC)[21] was used. 6 pmole trypSL (1.25 µl from a 50 µM stock was kinased with 2 µl 32PgammaATP (25 µM, 6000 Ci/mmol, 150 mCi/ml, Perkin Elmer) with 10 U PNK in 10 µl with 20 U RNasin (Roche). After 1 h, the probe was extracted by phenol/CHCl$_3$ and precipitated. The second phosphate was then added with Myokinase (Sigma M3003, Myokinase was dialyzed into 100 mM NaCl, 50 mM TrisHCl pH 7.5, 1 mM MgCl$_2$, 1 mM DTT), 100 U in 20 µl, in a total volume of 40 µl to 2.4 pmole trypSL in the presence of 1 mM ATP and 20 U RNasin (Roche) in vaccinia capping buffer. After 2 h, the RNA was extracted by phenol/CHCl$_3$ and precipitated. Capping was then done in 20 µl with vaccinia capping enzymes (NEB) according to the manufactures instructions and after 90 min 2 µl terminator nuclease buffer A and 0.7 U Terminator nuclease (Epicenter) were added. After 30 min, the RNA was extracted by phenol/CHCl$_3$ and precipitated. The RNA was then analyzed on 20% polyacrylamide gels, dried and exposed to a phosphoimager screen.

RNAse I digestion to analyze 2′-O-ribose methylation was done in the presence of 10 U T4 PNK (NEB) in 50 mM Tris-acetate (pH 6.5), 50 mM NaCl, 10 mM MgCl$_2$ and 2 mM DTT to remove 2′,3′-cyclic phosphate intermediates[69].

For RNA stability experiments, 32P labeled uncapped and capped trypanosome splice leader oligo with or without cOMe was incubated in a total volume of 10 µl, in 40% (v/v) nuclear or cytoplasmic extract, 1 mM ATP, 5 mM creatine phosphate, 2 mM MgAcetate, 20 mM KGlutamate, 1 mM, DTT, 20 U RNasin (Roche), and 5 µg/mL tRNA on ice for 45 min. Input was taken before the addition of nuclear extract. The RNA was extracted by phenol/CHCl$_3$ and precipitated. Samples were then separated on 8% polyacrylamide gels, dried and exposed to a phosphoimager screen.

For immunoprecipitations of CBP80, 32P labeled capped Trypanosome splice leader oligo with or without cOMe was incubated in a final volume of 120 µl in IP-Buffer (150 mM NaCl, 50 mM Tris HCL, pH 7.5, 1% NP-40, 5% glycerol) together with nuclear extract (40%, v/v), rabbit anti-CBP80 (4 µl, gift from D. Kopytova), 20 µl protein A/G beads (SantaCruz) in the presence of Complete Protein Inhibitor (Roche) and 40 U RNase inhibitors (Roche) for 2 h at 4 °C. After washing the beads, RNA was extracted by phenol/CHCl$_3$ and precipitated. Samples were then separated on 8% polyacrylamide gels, dried, and exposed to a phosphoimager screen.

**Substrate RNAs, S2 cell expression of CMTrs, and in vitro 2′-O-ribose methylation**. Substrate RNAs for in vitro methylation assays were 5′UTRs from either the per (31 nts) or Sdt (28 nts) cut and cloned in to a modified pBS cut with Kpn and Xho with primers per T7 2.5A (CAGTAATACGACTCACTATTAGTG TTCGTGCGAATTTAGAGCCAGAAGGTC), per T7 2.5B (TCGAGACCT TCTGGCTCTAAATTCGCACGAACACTAATAGTGAGTCGTATTACTGG TAC), and Sdt T7 2.5A (CCAGTAATACGACTCACTATTAGTTGCGAGGCC GACCGTCGACGTTTCC) and Sdt T7 2.5B (TCGAGGAAAACGTCGACGG TCGGCCTCGCAACTAATAGTGAGTCGTATTACTGGGTAC) containing a T7 phi2.5 promoter[70]. Plasmids were linearized with Xho and EcoRV, phenol/CHCl$_3$ extracted and ethanol precipitated in the presence of glycogen. In vitro transcription was done from 1 µg linearized plasmid with T7 MegaScript kit (Ambion) according to the manufacturer's instructions. For an in vitro transcript starting with G, pBS SK+ was linearized with MseI to yield the sequence (GGGCGAAT TGGGTACGATCCTCTAGCCAACAATT). In vitro transcripts were analyzed by dephosphorylation and labeling with 32PgammaATP on 20% denaturing poly-acrylamide gels alongside appropriate markers for the correct size and after digestion with nuclease P1 on 2D TLCs for correct transcription initiation.

RNAs (2 µM) were capped in 20 µl with 6 µl [α-32P] GTP (3000 Ci/mmol, 6.6 µM; Hartmann Analytics, Germany) using vaccinia capping enzyme in the

presence of SAM (2 mM) and 0.5 μl RNAse protector (Roche) for 1 h at 37 °C according to the manufacturer's instructions (NEB).

For expression of dCMTrs, *Drosophila* S2 cells were transfected as described[39]. Cells were then pelleted and homogenized in cell lysis buffer (20 mM TrisHCl pH 8, 137 mM NaCl, 1 mM EDTA, 25% glycerol (Ultrapure), 1 mM DTT, 1% NP40) in the presence of protease inhibitor cocktail (Roche) and 2% PMSF (200 mM in isopropanol) added step-wise. 100 μl extract was made 48 h after transfection of two 6-well plates, and 50 % extract was used in 10 μl methylation assays with 50% capping buffer (NEB) mix containing SAM (4 mM), 2 mM MgCl₂, 1 μl RNase protector (Roche) and substrate RNA, and incubated for 30 min at room temperature. The RNA was then purified with AMPure XP magnetic beads (Beckman Coulter) and digested with RNase I as described before.

**Immunostaining of tissues, RNA in situ hybridization, and puromycin labeling.** In situ antibody stainings were done as described previously[37] and below using rat anti-HA (MAb 3F10, 1:20; Roche), rabbit anti-FLAG (M2, 1:250, SIGMA), mouse anti-ELAV (MAb 7D, 1:20, which recognizes 7 amino acids unique to ELAV) and anti-GFP (1:250; Invitrogen A11122) and visualized with Alexa Fluor 488- and/or Alexa Fluor 647-coupled secondary antibodies (1:250; Molecular Probes or Invitrogen, A11034). DAPI (4′,6-diamidino-2-phenylindole) was used at 1 μg/ml. For imaging, tissues were mounted in Vectashield (Vector Labs) for confocal microscopy using a Leica TCS SP5/SP2 and LAS-X software. Images were processed using Fiji 1.53c.

To analyze synapses at NMJ third instar wandering larvae of third chromosome inserts for *UAS Y14 HA:FLAG*[71], *UAS elF4AIII:HA:FLAG*[71], *UAS CBP20:HA* (FlyORF) and *UAS elF4E:HA* (FlyORF) crossed to *elav^C155-GAL4* were dissected in PBS and fixed with Bouin's solution (Sigma-Aldrich, HT10132) for 5 min. The samples were washed three times in PBT (PBS with 0.1% Triton™ X-100 (Sigma, T8787) and 0.2% BSA) for 15 minutes. Primary antibody were rat anti-HA (MAb 3F10 1:20, Roche) or rabbit anti-FLAG (M2, 1:250, SIGMA), Mouse anti-NC82 (1:50, DSHB), rabbit anti-CBP80 (1:100, Gift from D. Kopytova)[72] and DAPI (4 = 6 = -diamidino-2-phenylindole,1 μg/ml) was carried out overnight at 4 °C followed by secondary antibodies (conjugated with Alexa Fluor 488 or Alexa Fluor 647 (1:250; Molecular Probes, Invitrogen) at RT for 4–5 h. All antibodies used have been validated by in situ staining or Western blots. NMJs were mounted in Vectashield (Vector Labs), scanned with Lecia TCS SP8, and processed using FIJI. For quantification of synapse staining, the mean intensity of the boutons was calculated using the Nikon NIS-Elements Basic Research (BR) imagining software, and the data were analyzed using GraphPad Prism.

For RNA in situ hybridizations to polyA mRNA, a 45 nt rhodamine-labeled oligo(dT) oligonucleotide was used[73]. After fixation in 4% paraformaldehyde in PBS for 15 min at room temperature, samples were washed for 30 min with PBS, 0.1% TritonX-100, and then incubated for 20 min in 2× SSC, 0.1% TritonX-100. Samples were then incubated in pre-hybridization buffer (25% formamide, 2X SSC, 0.1% TritonX-100) before hybridization (25% formamide, 2× SSC pH 7.2, 10% w/v dextran sulfate (Sigma Aldrich), 1 mg/mL E. coli tRNA (Sigma-Aldrich, R1753), 0.1% TritonX-100) for 48 h at 42 °C. Samples were first washed in 50% Formamide, 2× SSC, 0.1% TritonX-100 at room temperature for 1 h, then three washes in 2× SSC, 0.1% TritonX-100 followed by 1× SSC, 0.1% TritonX-100 for 10 min each at 42 °C. Samples were then stained with antibodies and analyzed by confocal microscopy as described.

For the analysis of nascent protein synthesis at synapses, pinned third instar larval fillets were incubated in Insect Express medium (Lonza) containing 1 mM puromycin for 30 min. Fillets were then fixed as above and stained with anti-puromycin (mouse monoclonal 3RH11 1:200, Millipore) and anti-HRP (1:250, 323 005 021, Jackson ImmunoResearch) and secondary antibodies as described above.

**Statistical analysis of NMJ's, RNA stability, IP's, and puromycin incorporation.** Data were analyzed using GraphPad Prism 7. Two-tailed *t* tests were used for comparing two groups, and one-way ANOVA followed by a Tukey's post hoc test was used for comparing multiple groups.

**Polytene chromosome preparations and stainings.** CMTr1 and CMTr2 were expressed in salivary glands with *elav^C155-GAL4* from a UAS transgene tagged with HA or FLAG, respectively, as described[39]. Briefly, larvae were grown at 18 °C under non-crowded conditions. Salivary glands were dissected in PBS containing 4% formaldehyde and 1% TritonX100, and fixed for 5 min, and then for another 2 min in 50% acetic acid containing 4% formaldehyde, before placing them in lactoacetic acid (lactic acid:water:acetic acid, 1:2:3). Chromosomes were then spread under a siliconized coverslip and the coverslip was removed after freezing. Chromosome was blocked in PBT containing 0.2% BSA and 5% goat serum and sequentially incubated with primary antibodies (mouse anti-PolII H5 IgM, 1:1000, Abcam, and rat anti-HA MAb 3F10, 1:50, Roche, or rabbit anti-FLAG, 1:1000, SIGMA) followed by incubation with Alexa488- and/or Alexa647-coupled secondary antibodies (Molecular Probes) including DAPI (1 μg/ml, Sigma).

**Illumina sequencing and analysis of differential gene expression.** For sequencing, QuantSeq 3′ FWD libraries were generated from *y w* control and *y w CMTr^13A*; *CMTr^M32* flies. The QuantSeq 3′ FWD kit was used according to the

manufacturer's instructions with the following modifications: RNA was not denatured, and 6 U of Heparinase I (NEB) was added to the first-strand cDNA synthesis mix. Pooled indexed libraries were sequenced on an Illumina NextSeq 500 to yield between 10 and 30 million single-end 50 bp reads per sample.

After demultiplexing with Illumina bcl2fastq v1.8.4, sequence reads were aligned to the Drosophila genome (dmel r6.02) using STAR 2.6. Reads for each gene were counted using HTSeq-count and differential gene expression determined with DESeq2 and the Benjamini-Hochberg for multiple testing to raw *P*-values with *p* < 0.05 considered significant.

**CLIP of CMTr targets.** For CLIP, RNA was prepared essentially as described from 14 to 18 h old embryos of *y w CMTr^13A*; *gCMTr1:TEVHA CMTr^M32 gCMTr2:TEVFLAG*[74]. Embryos were first dechorionated in 50% bleach, washed and then fixed in heptane containing 5% formaldehyde (10 ml heptane, 1.75 ml 37% formaldehyde, and 1.3 ml PBS equilibrated for 30 min) for 10 min with vigorous shaking. Embryo extracts were then prepared in RIPA buffer (150 mM NaCl, 50 mM Tris–HCL, pH 7.5, 1% NP-40, 0.5% Na-deoxycholate, 0.05% SDS) in a 1-ml Dounce homogenizer. After 20–40 strokes with the tight pestil, 1 vol of immuno-precipitation (IP) buffer was added (150 mM NaCl, 50 mM Tris–HCL, pH 7.5, 0.05% NP-40). The extract was then cleared by centrifugation for 15 s. IPs were done with monoclonal anti-HA antibodies coupled beads (Sigma) or anti-FLAG antibodies and protein A/G beads (SantaCruz) in IP buffer containing 7 mM CaCl₂, 40 U of RNase inhibitor (Roche), 2 U of TurboDNase (Ambion), and 15% of extract for 2 hr at room temperature. After washing and TEV Proteinase (Promega) digestion for 1 h on ice, the supernatant was taken off, Proteinase K digested (0.5 mg/ml in 150 mM NaCl, 100 mM Tris–HCl, pH 7.5, 10 mM EDTA, 0.25% SDS) for 30 min at 37 °C, and RNA was isolated by phenol/chloroform extraction and ethanol precipitation in the presence of glycogen.

The RNA was then reverse-transcribed with Superscript II (Invitrogen) according to the manufacturer's instructions using a random nonamer tagged partial p7 sequence (CACGACGGCTCTTCCGATCTNNNNNNNNN) and the first-strand synthesis product was purified using AMPure XP beads (Beckman) following the manufacturer's instructions with 1.8 volumes. To generate double-stranded cDNA and sequencing-ready libraries, Lexogen's quant-seq 3′ FWD kit was used, proceeding from the RNA removal and second-strand synthesis steps. The input library was generated the same way from RNA before IP. Size selection of libraries was carried out with PAGE prior to sequencing with the NextSeq 500. Differential gene expression analysis performed as above then provided a simple route to detecting enriched transcripts following immuno-precipitation.

**Data analysis.** GO enrichment analysis was performed with Pantherdb. Gene expression data were obtained from flybase. Visualization of RNA-seq data was carried out with the R packages EnhancedVolcano Version 1.4.0, STAR 2.7.9a, and ggplot2 in the R studio environment[75,76].

Hypergeometric *p*-values for the significance of overlapping genes between CMTr2 and FMRF CLIP targets were calculated using the 197 successes in a sample size of 701 CMTR2 clip targets, compared to the 2432 successes of FMRP targets in cholinergic and GABAergic neurons[50] in the whole population of 17,421 coding genes that can be returned following alignment ($p = 1.34^{-23}$).

**Reporting summary.** Further information on research design is available in the Nature Research Reporting Summary linked to this article.

## Data availability
The data supporting the findings of this study are available from the corresponding authors upon reasonable request. Gene expression data have been deposited at GEO under the accession numbers GSE116212, GSE181321, and GSE138868. Source data are provided with this paper as a Source Data file. Source data are provided with this paper.

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

## Acknowledgements

We thank S. Brogna, C. Samakovlis, B. Suter, J.-Y. Roignant, M. Ramaswami, FlyORF and the Bloomington and Kyoto stock centers for fly lines, S. Brogna, D. Kopytova, P. Lasko and the Developmental Studies Hybridoma Bank for antibodies, S. Brogna for an oligo(dT) probe, the University of Cambridge Department of Genetics Fly Facility for injections, BacPac for DNA clones, E. Zaharieva for help with stainings, D. Balacco and F. Stappers for help with artwork, and B. Muller and R. Michell for comments on the manuscript. MS is funded by the BBSRC (BB/R002932/1) and the Leverhulme Trust, RGF from BBSRC (BB/R001715/1), DH from WISB, a BBSRC/EPSRC Synthetic Biology Research Centre (BB/M017982/1) and BBSRC (BB/L006340/1), and S.W. by a Wellcome Principal Research Fellowship (200846/Z/16/Z) and an ERC Advanced Grant (789274). NA is a Nottingham Research Fellow funded by the University of Nottingham.

## Author contributions

I.U.H. and M.S. performed biochemistry, molecular biology, and genetic experiments, Y.W. and S.W. performed learning experiments and anatomical analysis of adult brains, M.P.N. performed antibody stainings, N.A., Z.B. and R.F. performed sequencing and biochemistry experiments. N.A. and D.H. analyzed sequencing data. I.U.H. and M.S. conceived the project and wrote the original draft of the paper. S.W., R.F., Z.B., N.A. D.H., and all other authors reviewed and edited. M.S., S.W., R.F. and D.H. supervised and acquired funding.

## Competing interests

The authors declare no competing interests.
