## [Peer Review File · Nature Communications]

Title: CMTr cap-adjacent 2'-O-ribose mRNA methyltransferases are required for reward learning and mRNA localization to synapsesREVIEWER COMMENTS

Reviewer #1 (Remarks to the Author):

The paper by Hausmann et al used the fly genetics to characterize 2 cap methyltransferases (CMTr), Cmtr1 and Cmtr2/adrift in the fly. The authors reported that both CMTr1 and CMTr2 can methylate the 2'-OH position of the ribose of the first nucleotide in mRNA (cOMe). Flies lacking both CMTs are viable and show defective rewarding learning through the function of CMTs in mushroom body neurons. The RNA sequencing data identified altered expression in several cell adhesion and signaling molecules. The authors found that increased eIF4E and decreased CBP20, CBP80, Y14 and eIFIII A at the presynaptic sites of neuronal muscular junctions (NMJs) and proposed that CMTr1/2-mediated cap methylation is important to transport RNAs through the interaction with the CBP complex. Their findings are very surprisingly because the corresponding CMTr1 and CMTr2 in mammals have unique function to catalyze cOMe in the first and second nucleotide of mRNA, respectively. The studies by Pelletier (J Biol Chem 285: 33037) and Bujnicki groups (Nucl. Acids Res 39:4756) clearly demonstrated that both CMTr proteins cannot substitute the function for each other in vitro. Thus, CMTr1 is known for cap1 (M7GpppNmN) methylation, while CMTr2 is known for cap2 (M7GpppNNm) methylation. In line with the in vitro study, a recent study by Huang group (Cell Rep 33:108269) also reported that CMTr1-knockout mice are embryonic lethal and cap1 methylation is completely abolished in CMTr1-knockout neurons. Although Hausmann et al. study reported many aspects of CMTr1 and CMTr2 in the fly, their data on behavioral characterization, target mRNA identification and molecular mechanism, either standing alone or linking together, are relatively weak. Moreover, what is the basis to define the 2 fly CMTr genes equivalent to mammalian CMTr1 and CMTr2? The authors only checked the cap1 but not cap2 methylation in CMTr2-mutant flies, so will it be possible that Drosophila has 2 cap1 methyltransferases but has no cap2 methyltransferase or Drosophila CMTr2 can catalyze cOMe in the first 2 nucleotides of mRNA? Such a fundamental question should be clearly addressed to help understanding the molecular and physiological functions of cap methylation in the fly.

Specific comments:

1. Both CMTr1 and CMTr2 mutant flies are deletion mutants. Do the authors know whether truncated CMTr1 and CMTr2 RNAs and proteins are expressed in the mutant flies? Moreover, how are the domain structure and similarity in fly CMTs compared to those in mammalian CMTs? The data to show the disappearance of cap1 methylation came from TLC results in Fig. 1g-1k. Nevertheless, I am very confused about how the authors can get the quantified result in Fig. 1l, because pCm and pGm signals are not found in the TLC shown in Fig. 1. Moreover, pUm is found in Fig. 1i CMTr1 mutant fly (Fig. 1i) and CMTr2 mutant fly (Fig. 1j) but was mysteriously absent in wild-type fly (Fig. 1h). The TLC position showing whether the pUm signal should be for double mutant fly has been cropped (Fig. 1k), so an uncropped TLC image should be replaced here. Because rRNA is also 2'-O-ribose methylated, are poly(A) RNAs used here completely depleted of rRNA? How sure is the signal in TLC plates from cOMe instead of internal OMe in mRNA and rRNA?

2. Distribution of CMTr2 to the cell membrane is an interesting finding (Fig S1). Capping of RNA is known

to occur co-transcriptionally in the nucleus (nascent mRNA) and likely also in the cytoplasm (recapping of mRNA). However, it is not clear whether cap1 methylation can happen in the cytoplasm. 2In CMTr1 mutant flies, CMTr2 could compensate its role for cap1 methylation. Under such a condition, does it affect the subcellular distribution of CMTr2, more prominently present in the nucleus than cytoplasm and cell membrane? Using biochemical fractionation and western blotting to detect the subcellular amount of CMTr2-flag in CMTr1 mutant fly could be an alternative approach.

3. For the rescue experiment in double mutant flies (Fig. 2c), please include catalytic-dead CMTr1 and CMTr2 mutant (mutating the critical residue in the catalytic triad) for single or double rescue. If the loss of cOME is responsible for reward learning deficit instead of other unknown function of fly CMTrs, the defective learning and memory in double mutant fly will not be rescued by expressing catalytically dead CMTr mutants. In the absence of these data, the authors should not conclude that “cOME deficiency specifically impairs reward learning”.

4. For Fig. 3, the result subtitle “CMTr loss increases transcript abundance” is misunderstanding. Is the total amount of mRNAs increased in double mutant fly or just select transcripts? Can the authors also perform the rescue with wild-type or catalytically dead CMTr1, CMTr2 or both to determine whether some identified mRNAs in cell adhesion and signaling related to learning and memory can be restored to the normal level with methyltransferase-intact CMTrs? Although cOME appears in all mRNAs, the level of most mRNAs remained unchanged (Fig. 3a). Thus, using in vitro synthesized cap-methylated reporters (Fig. S4) to conclude that cOME deficiency does not affect mRNA stability may be incorrect. Perhaps, the stability of some downregulated transcripts in Fig. 3a may be reduced in double mutant flies.

5. In Fig. 4, unlike CMTr1, CMTr2 seems to regulate a subset of mRNAs, so the authors performed CLIP for CMTr1-HA and CMTr2-flag in double mutant fly. Surprisingly, 36 and 701 targets were identified for CMTr1-HA and CMTr2-flag, respectively, though CMTr1-HA is associated with RNA polymerase II more than CMTr2-flag in polytene chromosomes (Fig. 4a-j). So, why did CLIP identify more targets associated with CMTr2 but not CMTr1? Are the authors sure about that no transcript expression is altered in single CMTr-defective fly? Moreover, it will be interesting to learn whether the amount of CMTr2-flag in polytene chromosomes increases in CMTr1-mutant fly and whether CMTr2 can compensate the loss of CMTr1 to methylate mRNAs cotranscriptionally in the nucleus.

6. CBP20 instead of CBP80 is the subunit of CBP complex directly binding to m7G cap. The structure of CBP20/CBP80 in complex with or without m7GpppG has been solved (Nat Struct Biol 9:912). Moreover, another study (RNA 11:1355) showed that CBP20/CBP80 binds similarly to m7GpppG and m7GpppGm and slightly tighter to m7GpppAm than m7GpppA. Can the authors perform structural remodeling to demonstrate whether the cap-binding pocket of fly CBP20 can recognize not only the m7G motif but also cOME of the first nucleotide?

7. Up to current literatures, there is no evidence showing that CMTr1 could methylate 2nd and 3rd nucleotides of mRNA. Hence, in order to prove that cOME of the first nucleotide by CMTr1 and CMTr2 affects CBP binding, the mRNA substrate used for Fig. 5a should be cap1 RNA. Alternatively, the authors

could immunoprecipitate the CBP complex from wild-type and double mutant flies and identify which mRNAs show defective binding to the CBP complex.

8. In Fig. 5, the authors showed that decreased CBP20, CBP80, Y14 and eIFIII A at the presynaptic sites of NMJs. However, if I understand correctly, the transcriptomic result (Fig. 3) was obtained by using RNAs extracted from the whole flies. CLIP was performed by using elav-GAL4 (neuron) driven epitope-tagged CMTrs (Fig. 4). Defective reward learning in double mutant flies is mainly caused by the deficiency of CMTrs in mushroom Kenyon cells (Fig. 2), yet the possible molecular mechanism was examined in NMJs (Fig. 5). To be able to compare the results from behavior, transcriptome, immunostaining and molecular assays, will it be more cohesive to focus on Kenyon cells instead of NMJs? Alternatively, according to the authors, the evident change of presynaptic CBP complex, EJC components and eIF4E at NMJs likely affect localization of mRNAs and local translation. If so, how about the number and size of synaptic boutons of NMJs and the crawling behavior of larvae?

9. It is an interesting hypothesis that CMTr-mediated cOMe facilitates transport and localization of mRNA to distal synapses. However, more rigorous tests and stringent controls are required prior to reaching this conclusion. Since CMTr1 is mainly localized in the nucleus and cap1 methylation occurs co-transcriptionally, the authors could perform rescue with different forms of CMTr1 and CMTr2, such as catalytically dead and nuclear localization-defective mutants, to at least demonstrate that cap1 methylation of mRNAs in the nucleus is required for transport and synaptic localization. Can the authors at least select one or two targets from the transcriptome and CLIP data whose transport and translation are affected in double mutant flies to affect synaptic plasticity?

Reviewer #2 (Remarks to the Author):

The manuscript by Hausmann et al. presents the characterization of two enzymes in *Drosophila* required for 2'-O-methylation of the ribose of the first nucleotide in mRNA. A loss of function for the two genes encoding these enzymes was carried out and the mutants were further analysed. Surprisingly the lack of these two genes does not alter development or viability but the flies exhibit a very specific behavioral defect in reward learning. The authors further went on to identify potential targets using a combination of transcriptomic and CLIP-seq approaches. Molecularly this modification does not seem to alter transcript stability but instead seems to be required for preventing premature translation in axons.

Overall this is a very interesting study that provides novel insights into the function of cOMe modification, which remains poorly understood and so far has been mostly studied in cell culture and in the context of immune function. The authors use an elegant combination of genetic, molecular, transcriptomic and imaging approaches to uncover the role of this modification in vivo. While there are still some questions that arise from this work and are not yet fully addressed I believe that the main conclusion of this work is original and must be made available to the community.

I have only a few comments/suggestions that should help strengthening the main conclusions.

Comments:

L35-36: The functions of cOMe in animals, however, remain unknown". This is not entirely true. There are some recent articles demonstrating the function of CMTR1 in neuronal development in mouse, and in the RNAi pathway in *Drosophila*.

The introduction is very short and could be expanded. In particular more details about the known functions of cOMe could be included (e.g. role in self versus non self recognition, role in the RNAi pathway). Also, the target specificity of these enzymes could be better introduced. For instance CMTR2 is known to methylate the ribose of the second nucleotide in vertebrates, which seems different in *Drosophila* but this was never discussed. Also, more introduction about reward learning and other types of learning would help to understand the main data.

Fig 1a-c. The names of the primers used in 1c could be indicated in 1b and c.

L54: "Dynamic O-methylation at the 2' position of the ribose...". In what sense this a dynamic modification? Please explain.

L83-86: What does this sentence mean? It seems that a conclusion is missing. Are the genes carrying the INR promoter sequence more susceptible to be methylated?

Sentences in L103 and 104 are somewhat redundant.

L161-162: In what sense the effect is different from the loss of m6A? Please explain

L164-165: What is the proportion of cOMe in *Drosophila* and mouse? The reference to fig 1e and f is not correct.

L165-L166: This conclusion may be tuned down since a direct test to address this question was not performed (e.g infection with virus). Furthermore cOMe seems to play a role in the RNAi pathway and may therefore have some immune functions in *Drosophila* too.

The experiment in Fig. 3C is a bit hard to interpret since in this in vitro system the first three nucleotides are methylated, which is probably different in *Drosophila*. I suggest to mention this potential caveat.

L184: the reference to Fig 1e and f is wrong.

The results of the CLIP experiment is counter intuitive with regards to the staining of the polytene chromosomes. The explanation provided in L197-198 is not very satisfying. If CMTr1 associates with

most genes this should be reflected in the CLIP data, unless CMTr1 does not bind RNA directly. Could the authors show representative tracks of the CLIP data? Is the binding always occurring at 5' end? If not, could they classify binding with respect to 5'UTR, CDS and 3'UTR?

The model drawn from the immunostaining data in Fig 5 is that less CBP staining in CMTr1/2 mutants results in premature translation and therefore less localized untranslated transcripts at synapses. I wonder whether the decrease in CBP staining is only observed at synapses or whether it already happens in the nucleus? If so, the main effect could be defective mRNA export (as capping is required for this process), rather than premature translation. I wonder whether the authors could devise an approach to monitor nascent translation at NMJ (e.g. Trap-seq, fluorescence imaging)

Fig 5h: Is this overlap significant?

Reviewer #3 (Remarks to the Author):

In this manuscript, the authors have investigated the role that methylation of mRNAs that are O-methylated at the 2' position of the ribose (cOMe). The authors show that the two cap methyltransferases (CMTr1 and CMTr2) of *Drosophila* are able to methylate the ribose of the first nucleotide in mRNA. Surprisingly, flies with deletion of CMTr1 and CMTr2 did not have cOMe, but they are viable. The authors went to perform behavioral studies and found that double mutant flies have a reward learning defect that can be rescued by conditional expression in mushroom body neurons before training. The authors also performed RNA sequencing on control flies and flies that lack cOMe and found a number of genes with altered expression. Interestingly, the majority of genes altered in the cOMe-lacking flies were upregulated, which argues against the proposed role of cOMe protecting mRNAs from degradation. The authors also performed CLIP-seq and found that CMTr2 binds to transcripts/targets associated with cell adhesion and signaling relevant for learning that are also targets. Finally, the authors show that cOMe is required for localization of untranslated mRNAs to synapses and enhances binding of the cap-binding complex in the nucleus.

The authors present a number of interesting findings in the manuscript that will be of interest to a wide variety of RNA biologists and neuroscientists. The manuscript is clearly written and concise. The data presented appear to be of good quality. I have a couple of comments for the authors to consider.

1) One of the main conclusions of the authors is that the role of cOMe is to co-transcriptionally prime mRNAs for localized protein synthesis at synapses, but they never directly test this. One thing that the authors could do is determine whether general protein synthesis is altered in either the soma or dendrites of cOMe-lacking flies. This could be done with either fluorescent SUNSET or FUNCAT.

2) Similarly, can the authors show that local synthesis of one of the CMTr2 targets is actually altered? A proximity ligase assay (PLA) could be used for this where a puromycin antibody is used in conjunction to

an antibody against the CMTr2 target picked.

3) If the authors cannot do a SUNSET, it would important to demonstrate that at least one or two of the mRNAs that lack cOMe are localized improperly.

Point by point response to reviewers' comments

We would like to thank the reviewers for carefully reading and critically assessing our work, and their constructive comments in support for publication of our findings about the molecular genetic characterization of CMTrs in a *Drosophila* model. We are very pleased to hear that the reviewers find our discoveries exciting with excellent supporting analysis.

Reviewer #1 (Remarks to the Author):

The paper by Haussmann et al used the fly genetics to characterize 2 cap methyltransferases (CMTr), Cmtr1 and Cmtr2/adrift in the fly. The authors reported that both CMTr1 and CMTr2 can methylate the 2'-OH position of the ribose of the first nucleotide in mRNA (cOMe). Flies lacking both CMTrs are viable and show defective rewarding learning through the function of CMTrs in mushroom body neurons. The RNA sequencing data identified altered expression in several cell adhesion and signaling molecules. The authors found that increased eIF4E and decreased CBP20, CBP80, Y14 and eIFIII A at the presynaptic sites of neuronal muscular junctions (NMJs) and proposed that CMTr1/2-mediated cap methylation is important to transport RNAs through the interaction with the CBP complex. Their findings are very surprisingly because the corresponding CMTr1 and CMTr2 in mammals have unique function to catalyze cOMe in the first and second nucleotide of mRNA, respectively. The studies by Pelletier (J Biol Chem 285: 33037) and Bujnicki groups (Nucl. Acids Res 39:4756) clearly demonstrated that both CMTr proteins cannot substitute the function for each other in vitro. Thus, CMTr1 is known for cap1 (M7GpppNmN) methylation, while CMTr2 is known for cap2 (M7GpppNNm) methylation. In line with the in vitro study, a recent study by Huang group (Cell Rep 33:108269) also reported that CMTr1-knockout mice are embryonic lethal and cap1 methylation is completely abolished in CMTr1-knockout neurons. Although Haussmann et al. study reported many aspects of CMTr1 and CMTr2 in the fly, their data on behavioral characterization, target mRNA identification and molecular mechanism, either standing alone or linking together, are relatively weak. Moreover, what is the basis to define the 2 fly CMTr genes equivalent to mammalian CMTr1 and CMTr2? The authors only checked the cap1 but not cap2 methylation in CMTr2-mutant flies, so will it be possible that *Drosophila* has 2 cap1 methyltransferases but has no cap2 methyltransferase or *Drosophila* CMTr2 can catalyze cOMe in the first 2 nucleotides of mRNA? Such a fundamental question should be clearly addressed to help understanding the molecular and physiological functions of cap methylation in the fly.

We want to thank Referee No 1 for careful, critical and constructive evaluation of our manuscript and pointing out a key unresolved question, namely to unambiguously determine the methylation status of the second nucleotide.

We have now addressed this fundamental question whether *Drosophila* CMTrs can methylate both the 1st and/or 2nd nucleotide. We have developed an assay based on recapping specifically mRNA with ³²PalphaGTP and digestion with RNase I to determine the number of methylated nucleotides in *Drosophila* in vivo using mutant flies and in vitro with recombinant enzymes. Our data unambiguously show that both CMTrs in *Drosophila* can methylate the first nucleotide and that no methylation of the second nucleotide is detected.

Moreover, we have added an alignment of the annotated *Drosophila* orthologues of CMTr1 and CMTr2 to show that they have the same domain structure and are thus true equivalents of human CMTr1 and CMTr2.

Specific comments:

1. Both CMTr1 and CMTr2 mutant flies are deletion mutants. Do the authors know whether truncated CMTr1 and CMTr2 RNAs and proteins are expressed in the mutant flies?

The deletion in CMTr1 starts before the ORF and the remaining ORF does not encode a ATG, so no truncated protein is expressed. For CMTr2, the deletion starts after the ATG and results in a premature stop codon, so this allele potentially could express a 55 aa peptide, but no antibodies are available to test this.

Moreover, how are the domain structure and similarity in fly CMTrs compared to those in mammalian CMTrs?

We have added an alignment in Suppl Fig 1 of CMTr1 and CMTr2 to indicate that the annotated *Drosophila* orthologues have the same domain structure and are true equivalents of human CMTr1 and CMTr2.

The data to show the disappearance of cap1 methylation came from TLC results in Fig. 1g-1k. Nevertheless, I am very confused about how the authors can get the quantified result in Fig. 1l, because pCm and pGm signals are not found in the TLC shown in Fig. 1.

For the data shown in Fig 1o (previously 1l) we have quantified the TLC data from CMTr1 and CMTr2 double mutants that do not contain methylated nucleotides.

Moreover, pUm is found in Fig. 1i CMTr1 mutant fly (Fig. 1i) and CMTr2 mutant fly (Fig. 1j) but was mysteriously absent in wild-type fly (Fig. 1h). The TLC position showing whether the pUm signal should be for double mutant fly has been cropped (Fig. 1k), so an uncropped TLC image should be replaced here.

We have replaced the TLC Figures from flies (Fig 1k-n) with representative ones and also indicated that pUm runs at the same position as pT, that is a carry-over from oligo dT selection of mRNA.

Because rRNA is also 2'-O-ribose methylated, are poly(A) RNAs used here completely depleted of rRNA? How sure is the signal in TLC plates from cOMe instead of internal OMe in mRNA and rRNA?

cOMe is absent in polyA mRNA of CMTr double mutants, so we are sure that cOMe is from polyA mRNA and not any contamination. As detailed in the Methods section, we have used double oligo dT enrichment, or single enrichment followed by riboMinus treatment or removal of rRNA by 5'-3' XrnI, which all give comparable results.

2. Distribution of CMTr2 to the cell membrane is an interesting finding (Fig S1). Capping of RNA is known to occur co-transcriptionally in the nucleus (nascent mRNA) and likely also in the cytoplasm (recapping of mRNA). However, it is not clear whether cap1 methylation can happen in the cytoplasm. In CMTr1 mutant flies, CMTr2 could compensate its role for cap1 methylation. Under such a condition, does it affect the subcellular distribution of CMTr2, more prominently present in the nucleus than cytoplasm and cell membrane? Using biochemical fractionation and western blotting to detect the subcellular amount of CMTr2-flag in CMTr1 mutant fly could be an alternative approach.

Our behavioural data using negative geotaxis assays and appetitive olfactory conditioning clearly show that CMTrs can compensate each other. Also, we show that both are present in the nucleus and cytoplasm, and because of this, demonstrating cap methylation in the cytoplasm is difficult to address. Consistent with our findings, cOMe is not completely absent

in single mutants. Clearly, whether CMTRs can compensate each other for specific targets will be one of the key questions of the future, but will require considerable methodology development.

3. For the rescue experiment in double mutant flies (Fig. 2c), please include catalytic-dead CMTr1 and CMTr2 mutant (mutating the critical residue in the catalytic triad) for single or double rescue. If the loss of cOME is responsible for reward learning deficit instead of other unknown function of fly CMTRs, the defective learning and memory in double mutant fly will not be rescued by expressing catalytically dead CMTr mutants. In the absence of these data, the authors should not conclude that “cOME deficiency specifically impairs reward learning”.

Accordingly, we have changed the statement to “CMTr deficiency impairs reward learning” as already indicated in the title. The suggested experiments are interesting. However, these experiments involve making transgenes and complicated genetics including manipulation of 3 chromosomes, which will take more than a year and will not advance the central conclusion of the study.

4. For Fig. 3, the result subtitle “CMTr loss increases transcript abundance” is misunderstanding. Is the total amount of mRNAs increased in double mutant fly or just select transcripts?

We have amended this subtitle to say “CMTr loss increases abundance of certain transcripts”.

Can the authors also perform the rescue with wild-type or catalytically dead CMTr1, CMTr2 or both to determine whether some identified mRNAs in cell adhesion and signaling related to learning and memory can be restored to the normal level with methyltransferase-intact CMTRs? Although cOME appears in all mRNAs, the level of most mRNAs remained unchanged (Fig. 3a). Thus, using in vitro synthesized cap-methylated reporters (Fig. S4) to conclude that cOME deficiency does not affect mRNA stability may be incorrect. Perhaps, the stability of some downregulated transcripts in Fig. 3a may be reduced in double mutant flies.

We have amended our statement regarding mRNA stability to say that there could be sequence-specific effects on mRNA stability and that this needs to be addressed in follow up studies.

5. In Fig. 4, unlike CMTr1, CMTr2 seems to regulate a subset of mRNAs, so the authors performed CLIP for CMTr1-HA and CMTr2-flag in double mutant fly. Surprisingly, 36 and 701 targets were identified for CMTr1-HA and CMTr2-flag, respectively, though CMTr1-HA is associated with RNA polymerase II more than CMTr2-flag in polytene chromosomes (Fig. 4a-j). So, why did CLIP identify more targets associated with CMTr2 but not CMTr1? Are the authors sure about that no transcript expression is altered in single CMTr-defective fly?

We have repeated the CLIP of CMTr1 and the results are now drastically improved. Following deduplication by UMI and normalisation, we have over 1 million reads per replicate of the CMTr1 pull-down (1.2 and 1.9 Mio, respectively).

Moreover, it will be interesting to learn whether the amount of CMTr2-flag in polytene chromosomes increases in CMTr1-mutant fly and whether CMTr2 can compensate the loss of CMTr1 to methylate mRNAs cotranscriptionally in the nucleus.

CMTr1 is the main enzyme introducing cOME as measured by recapping leading to a 90% reduction in cOME levels (Fig 1g, lane 6) and thus CMTr2 does not generally compensate.

When we analysed polytene chromosomes for localization of CMTr2 expressed in a double knock-out we also did not see compensation. We have added these data in Suppl Fig 7 showing that CMTr2 staining does not expand in a global compensatory way.

6. CBP20 instead of CBP80 is the subunit of CBP complex directly binding to m7G cap. The structure of CBP20/CBP80 in complex with or without m7GpppG has been solved (Nat Struct Biol 9:912). Moreover, another study (RNA 11:1355) showed that CBP20/CBP80 binds similarly to m7GpppG and m7GpppGm and slightly tighter to m7GpppAm than m7GpppA. Can the authors perform structural remodeling to demonstrate whether the cap-binding pocket of fly CBP20 can recognize not only the m7G motif but also cOMe of the first nucleotide?

We have checked the structure of CBC bound to m7GpppG regarding contacts with cOMe. Although Tyr138 of CBC20 interacts with the first nucleotide through base-stacking and is in proximity to the 2' position, the main part bound by CBC is the m7G. Hence, it is likely that other proteins contribute to cOMe recognition.

7. Up to current literatures, there is no evidence showing that CMTr1 could methylate 2nd and 3rd nucleotides of mRNA. Hence, in order to prove that cOMe of the first nucleotide by CMTr1 and CMTr2 affects CBP binding, the mRNA substrate used for Fig. 5a should be cap1 RNA. Alternatively, the authors could immunoprecipitate the CBP complex from wild-type and double mutant flies and identify which mRNAs show defective binding to the CBP complex.

Our *in vivo* data clearly show that the cOMe is required for localization of the CBC bound to mRNA to synapses. However, as indicated by the structure of the CBC bound to m7GpppG, other proteins contribute to cOMe recognition. These studies require the generation of additional tools including antibodies against the CBC, as they are no longer available.

8. In Fig. 5, the authors showed that decreased CBP20, CBP80, Y14 and eIFIII A at the presynaptic sites of NMJs. However, if I understand correctly, the transcriptomic result (Fig. 3) was obtained by using RNAs extracted from the whole flies. CLIP was performed by using elav-GAL4 (neuron) driven epitope-tagged CMTrs (Fig. 4). Defective reward learning in double mutant flies is mainly caused by the deficiency of CMTrs in mushroom Kenyon cells (Fig. 2), yet the possible molecular mechanism was examined in NMJs (Fig. 5). To be able to compare the results from behavior, transcriptome, immunostaining and molecular assays, will it be more cohesive to focus on Kenyon cells instead of NMJs? Alternatively, according to the authors, the evident change of presynaptic CBP complex, EJC components and eIF4E at NMJs likely affect localization of mRNAs and local translation. If so, how about the number and size of synaptic boutons of NMJs and the crawling behavior of larvae?

We have added data to Fig 1f analysing the number of synapses at third instar NMJs in CMTr mutants to further support a role of cOMe at synapses. The negative geotaxis assay is an equivalent read-out to the study of larval crawling behaviour to reveal neuro-muscular impairment. Third instar NMJs are the classic model to study synapses in *Drosophila*, because of the stereotyped way *Drosophila* NMJs are built and because they are accessible. Projections of the kenyon cells are deep in the brain and synapses are not readily accessible. To be able to analyse synapses of Kenyon cells a battery of new tools would need to be developed. CLIP was done with genomic rescue constructs in a mutant background to be representative. We agree with this reviewer that ultimately the molecular basis of the learning defect needs to be explained, but this is far beyond the focus of this study.

9. It is an interesting hypothesis that CMTr-mediated cOMe facilitates transport and

localization of mRNA to distal synapses. However, more rigorous tests and stringent controls are required prior to reaching this conclusion. Since CMTr1 is mainly localized in the nucleus and cap1 methylation occurs co-transcriptionally, the authors could perform rescue with different forms of CMTr1 and CMTr2, such as catalytically dead and nuclear localization-defective mutants, to at least demonstrate that cap1 methylation of mRNAs in the nucleus is required for transport and synaptic localization. Can the authors at least select one or two targets from the transcriptome and CLIP data whose transport and translation are affected in double mutant flies to affect synaptic plasticity?

We have added data to Fig 6 demonstrating that CMTr-mediated cOMe is required for local translation at synapses. These data further support our findings shown in Fig 5 that CMTr-mediated cOMe is required for localization of untranslated mRNAs to synapses. We agree with this reviewer that elucidating the exact mechanism for synaptic localisation requires to identify specific targets. However, to study their localization sophisticated imaging techniques requiring tagging of cOMe targets with arrays of MS2 binding sites are a prerequisite. These are lengthy experiments involving complex genetics because of the redundancy of the two CMTrs that also localize to two different chromosomes.

Reviewer #2 (Remarks to the Author):

The manuscript by Haussmann et al. presents the characterization of two enzymes in *Drosophila* required for 2'-O-methylation of the ribose of the first nucleotide in mRNA. A loss of function for the two genes encoding these enzymes was carried out and the mutants were further analysed. Surprisingly the lack of these two genes does not alter development or viability but the flies exhibit a very specific behavioral defect in reward learning. The authors further went on to identify potential targets using a combination of transcriptomic and CLIP-seq approaches. Molecularly this modification does not seem to alter transcript stability but instead seems to be required for preventing premature translation in axons.

Overall this is a very interesting study that provides novel insights into the function of cOMe modification, which remains poorly understood and so far has been mostly studied in cell culture and in the context of immune function. The authors use an elegant combination of genetic, molecular, transcriptomic and imaging approaches to uncover the role of this modification in vivo. While there are still some questions that arise from this work and are not yet fully addressed I believe that the main conclusion of this work is original and must be made available to the community.

I have only a few comments/suggestions that should help strengthening the main conclusions.

We want to thank Referee No 2 for the positive feedback on our manuscript, appreciation of our comprehensive methodological approach and sharing the excitement about our new results.

Comments:

L35-36: The functions of cOMe in animals, however, remain unknown”. This is not entirely true. There are some recent articles demonstrating the function of CMTR1 in neuronal development in mouse, and in the RNAi pathway in Drosophila.

In light of these two recent publications we have adjusted this statement in the abstract.

The introduction is very short and could be expanded. In particular more details about the known functions of cOMe could be included (e.g. role in self versus non self recognition, role in the RNAi pathway). Also, the target specificity of these enzymes could be better introduced. For instance CMTR2 is known to methylate the ribose of the second nucleotide in vertebrates, which seems different in Drosophila but this was never discussed. Also, more introduction about reward learning and other types of learning would help to understand the main data.

We have expanded the introduction to include more background information relevant to our findings including the specificity of CMTrs including methylation of the first nucleotide by CMTr1 and the second by CMTr2 in humans. We further added information from two recent studies showing a knock-out of CMTr1 in mice and Drosophila with a minor role in the siRNA pathway.

To keep the introduction focused, additional information to explain the learning assays better and the role of cOMe in the immune system to the main text of the results.

In addition, a section to the discussion has been added addressing potential differences in target specificity of CMTRs between humans and Drosophila, but we also point out that methylation of the first nucleotide CMTr1 mutants in both mice and Drosophila is not completely absent.

Fig 1a-c. The names of the primers used in 1c could be indicated in 1b and c.

The primer names are now indicated in Fig 1a and b.

L54: “Dynamic O-methylation at the 2’ position if the ribose...”. In what sense this a dynamic modification? Please explain.

We have removed dynamic.

L83-86: What does this sentence mean? It seems that a conclusion is missing. Are the genes carrying the INR promoter sequence more susceptible to be methylated?

We have rephrased this sentence to clearly indicate that both methods to determine the first nucleotide in mRNA yield the same result.

Sentences in L103 and 104 are somewhat redundant.

We have removed the redundancy.

L161-162: In what sense the effect is different from the loss of m6A? Please explain

We have explained better that the complement of genes differentially expressed is different.

L164-165: What is the proportion of cOMe in Drosophila and mouse? The reference to fig 1e and f is not correct.

We have added to the text that cOMe in mice is in the low percent range and about 80% in flies, and we have corrected the reference to Fig 1g,h and j,k.

L165-L166: This conclusion may be tuned down since a direct test to address this question was no performed (e.g infection with virus). Furthermore cOMe seems to play a role in the RNAi pathway and may therefore have some immune functions in Drosophila too.

We have tuned down this statement.

The experiment in Fig. 3C is a bit hard to interpret since in this in vitro system the first three nucleotides are methylated, which is probably different in *Drosophila*. I suggest to mention this potential caveat.

We now mention this caveat.

L184: the reference to Fig 1e and f is wrong.

We have corrected this to Fig 11g,h and j,k.

The results of the CLIP experiment is counter intuitive with regards to the staining of the polytene chromosomes. The explanation provided in L197-198 is not very satisfying. If CMTr1 associates with most genes this should be reflected in the CLIP data, unless CMTr1 does not bind RNA directly. Could the authors show representative tracks of the CLIP data? Is the binding always occurring at 5' end? If not, could they classify binding with respect to 5'UTR, CDS and 3'UTR?

We have repeated the CMTR1 CLIP with much improved read counts. We included UMIs to enable PCR duplication removal, and present normalised, deduplicated reads in Data S3. These deduplicated and normalised reads total over 1 million reads per sample. Accordingly, we have amended the text to follow the data.

Due to experimental constraints, random fragmentation was not carried out prior to immunoprecipitation. Thus, detected RNAs were sequenced across the full length of the transcript and we would not expect significant 5' enrichment.

The model drawn from the immunostaining data in Fig 5 is that less CBP staining in CMTr1/2 mutants results in premature translation and therefore less localized untranslated transcripts at synapses. I wonder whether the decrease in CBP staining is only observed at synapses or whether it already happens in the nucleus? If so, the main effect could be defective mRNA export (as capping is required for this process), rather than premature translation. I wonder whether the authors could devise an approach to monitor nascent translation at NMJ (e.g. Trap-seq, fluorescence imaging)

We have added data to Fig 6 demonstrating that nascent translation at NMJs is reduced in the absence of cMOE.

Fig 5h: Is this overlap significant?

Yes, we have added a p-value to support this conclusion.

Reviewer #3 (Remarks to the Author):

In this manuscript, the authors have investigated the role that methylation of mRNAs that are O-methylated at the 2' position of the ribose (cOMe). The authors show that the two cap methyltransferases (CMTr1 and CMTr2) of *Drosophila* are able to methylate the ribose of the first nucleotide in mRNA. Surprisingly, flies with deletion of CMTr1 and CMTr2 did not have cOMe, but they are viable. The authors went to perform behavioral studies and found that double mutant flies have a reward learning defect that can be rescued by conditional expression in mushroom body neurons before training. The authors also performed RNA sequencing on control flies and flies that lack cOMe and found a number of genes with altered expression. Interestingly, the majority of genes altered in the cOMe-lacking flies were

upregulated, which argues against the proposed role of cOMe protecting mRNAs from degradation. The authors also performed CLIP-seq and found that CMTr2 binds to transcripts/targets associated with cell adhesion and signaling relevant for learning that are also targets. Finally, the authors show that cOMe is required for localization of untranslated mRNAs to synapses and enhances binding of the cap-binding complex in the nucleus.

The authors present a number of interesting findings in the manuscript that will be of interest to wide variety of RNA biologists and neuroscientists. The manuscript is clearly written and concise. The data presented appear to be of good quality. I have a couple of comments for the authors to consider.

We want to thank Referee No3 for careful evaluation and positive feedback on our manuscript and the recognition of a number of exciting findings of interest to a broad audience.

1) One of the main conclusions of the authors is that the role of cOMe is to co-transcriptionally prime mRNAs for localized protein synthesis at synapses, but they never directly test this. One thing that the authors could do is determine whether general protein synthesis is altered in either the soma or dendrites of cOMe-lacking flies. This could be done with either fluorescent SUnSET or FUNCAT.

We have added data to Fig 6 demonstrating that nascent translation at NMJs is reduced in the absence of cMOe using a SUnSET assay.

2) Similarly, can the authors show that local synthesis of one of the CMTr2 targets is actually altered? A proximity ligase assay (PLA) could be used for this where a puromycin antibody is used in conjunction to an antibody against the CMTr2 target picked.

Using a SUnSET assay, we show that local translation at synapses is reduced. For the suggested proximity ligation-assay we would need to separate cell bodies from synapses, but *Drosophila* is too small for this.

3) If the authors cannot do a SUnSET, it would be important to demonstrate that at least one or two of the mRNAs that lack cOMe are localized improperly.

We have established the SUnSET assay and demonstrate that local translation at synapses is reduced. We agree with this reviewer that there are a number of interesting follow up experiments building on our data. However, these experiments are lengthy and involve making several transgenes taking more than a year and thus more suitable for a follow up study.

REVIEWER COMMENTS

Reviewer #1 (Remarks to the Author):

I would like to thank the authors' effort to improve the manuscript by performing additional experiments. The new data clearly demonstrate that CMTr2 in the fly functions like CMTr1 for adding N1-cOMe on mRNA (Fig. 1g,1h, SuppFig 2). This is totally different from the role of mammalian CMTr2 in the literatures and is the first report to demonstrate some redundancy in CMTr1 and CMTr2. I still think it is important to perform the rescue with catalytic dead mutant in order to make a strong conclusion for the reason stated in the point #2. If the rescue experiment cannot be done during the limited time of revision, the authors should tone down "the lack of cOMe" as the cause for phenotypes.

Specific comments:

1. Just to fill in my knowledge gap, how does oligo-dT contamination generate such a strong background in TLC assay? I assume that an essential DNase treatment step was not included, so PNK labelling and nuclease P1 digestion can produce a lot of p-T. By checking the methodology, I found a mistake (likely). The authors mentioned "we decapped polyA mRNA with RppH and removed the first phosphate for labelling the first nucleotide by 32P.....". Can RppH, a bacterial enzyme, decap mRNA?

2. If the authors cannot provide the rescue experiment with catalytic dead CMTr1 and CMTr2, the Figure 2 conclusion should be changed to "CMTr1 and CMTr2 (instead of cOMe) are required for reward learning". After all, CMTr1 and CMTr2 are also cap-binding proteins, which may regulate neuronal function independent of their enzymatic activity. Especially from their own data, CMTr2-KO has less impact on N1-cOMe than CMTr1-KO (Fig. 1g), but more severe reduction in the number of synaptic boutons was observed in CMTr2-KO (Fig. 1f). Similarly, despite a great vs a small reduction of N1-cOMe in CMTr1 vs CMTr2 KO flies, the reward learning in both single KO is normal. Of course, CMTr1 and CMTr2 may regulate different target mRNAs; only when depleting their N1-cOMe together can impair reward learning. To some extent, overexpression of CMTr2 in dKO could rescue memory (Fig. 2d-e). Therefore, the reduced amount of N1-cOMe is not always correlated well with all phenotypes. If the rescue experiment cannot be done during the limited time of revision, the authors should tone down "the lack of cOMe" as the cause for phenotypes.

3. I asked the authors to focus on 1-2 target mRNA and demonstrate its reduced localization at synapses, because I do not expect that the absence of cOMe in mRNA would lead to a global downregulation of synaptic localization and translation of mRNAs. However, based on Fig 6 data, N1-cOMe is likely accounted for at least 40% synaptic translation. CBP20/80, Y14 and eIF4AIII are evidently reduced but eIF4E is increased (Fig. 5). The authors claimed not possible to demonstrate localization defect of a specific target mRNA but the experiment like in-situ hybridization to detect target mRNA instead of Ms2 tethered assay does not require the generation of new transgenic flies. Based on the new Fig 6a data- a dramatic reduction of new protein synthesis, I expect that in-situ signal of poly(A)-positive mRNAs should be significantly decreased in double-KO flies. Without the in-situ hybridization of poly(A)+ RNA, there is no data to demonstrate that "levels of untranslated mRNAs are reduced at synapses in the

absence of cOMe" (line 300)? If no specific target mRNA can be demonstrated in this study, they should at least compare the signal changes of puromycin labeling, poly(A)+ RNA (in situ hybridization), CBP20/80 and eIF4E between synapses and soma in wild-type and dKO flies. I suspect that the changes may also be observed in the soma. If so, the lack of CMTr1 and CMTr2 may just simply affect nuclear export instead of synaptic transport.

4. The author mentioned "In addition, cOMe is only in the low percent range in mice 9,40 and about 80% in Drosophila (Fig. 1g, h and j, k)9. For a primary role of cOMe for self/non-self discrimination one would expect 100% cOMe (line 209-211)." To our knowledge, N1-cOMe in mammals is detected in > 90 to 100% mRNA in several tissues and cell lines, so it is unclear why the authors said "low% in mice"?

Reviewer #2 (Remarks to the Author):

The authors addressed most of my comments and I congratulate them for the interesting work. There are several exciting questions for follow up studies, in particular regarding the specificity of the enzymes and the potential dynamicity of the modification.

Reviewer #3 (Remarks to the Author):

The authors have addressed my comments, including doing an experiment that was requested and adding the data to a figure in the paper. Congratulations on a very nice manuscript.

Point by point response to reviewers' comments

We are very pleased to hear that reviewers 2 and 3 have accepted our revisions and thank reviewer 1 for the positive evaluation of our exciting discoveries.

Reviewer #1 (Remarks to the Author):

I would like to thank the authors' effort to improve the manuscript by performing additional experiments. The new data clearly demonstrate that CMTr2 in the fly functions like CMTr1 for adding N1-cOMe on mRNA (Fig. 1g,1h, SuppFig 2). This is totally different from the role of mammalian CMTr2 in the literatures and is the first report to demonstrate some redundancy in CMTr1 and CMTr2. I still think it is important to perform the rescue with catalytic dead mutant in order to make a strong conclusion for the reason stated in the point #2. If the rescue experiment cannot be done during the limited time of revision, the authors should tone down "the lack of cOMe" as the cause for phenotypes.

We thank this reviewer for the positive evaluation of our work. We have addressed the additional comments and requests as detailed below including rephrasing to tone down conclusions to say "lack of CMTr's" rather than "lack of cOMe" as cause for phenotypes.

Specific comments:

1. Just to fill in my knowledge gap, how does oligo-dT contamination generate such a strong background in TLC assay? I assume that an essential DNase treatment step was not included, so PNK labelling and nuclease P1 digestion can produce a lot of p-T. By checking the methodology, I found a mistake (likely). The authors mentioned "we decapped polyA mRNA with RppH and removed the first phosphate for labelling the first nucleotide by 32P.....". Can RppH, a bacterial enzyme, decap mRNA?

RppH was the follow up enzyme for TAP (Tobacco acid pyrophosphatase), which was discontinued by Epicentre. NEB and others including us have shown that RppH works comparably to TAP.

The minor dT carry-over was only discovered because we used CMTr1 and CMTr2 double knockouts, which completely lack cOMe. We are currently working on a protocol to deal with this issue in TLCs.

2. If the authors cannot provide the rescue experiment with catalytic dead CMTr1 and CMTr2, the Figure 2 conclusion should be changed to "CMTr1 and CMTr2 (instead of cOMe) are required for reward learning". After all, CMTr1 and CMTr2 are also cap-binding proteins, which may regulate neuronal function independent of their enzymatic activity. Especially from their own data, CMTr2-KO has less impact on N1-cOMe than CMTr1-KO (Fig. 1g), but more severe reduction in the number of synaptic boutons was observed in CMTr2-KO (Fig. 1f). Similarly, despite a great vs a small reduction of N1-cOMe in CMTr1 vs CMTr2 KO flies, the reward learning in both single KO is normal. Of course, CMTr1 and CMTr2 may regulate different target mRNAs; only when depleting their N1-cOMe together can impair reward learning. To some extent, overexpression of CMTr2 in dKO could rescue memory (Fig. 2d-e). Therefore, the reduced amount of N1-cOMe is not always correlated well with all phenotypes. If the rescue experiment cannot be done during the limited time of revision, the authors should tone down "the lack of cOMe" as the cause for phenotypes.

We completely agree with this reviewer about the experiment with catalytically dead CMTrs and we have started to make these lines. However, since generating these lines in very lengthy we will use them in future studies. According to the reviewers suggestion we have amended the text to say "lack of CMTr" rather than "lack of cOMe" as cause for phenotypes.

3. I asked the authors to focus on 1-2 target mRNA and demonstrate its reduced localization at synapses, because I do not expect that the absence of cOMe in mRNA would lead to a global downregulation of synaptic localization and translation of mRNAs. However, based on Fig 6 data, N1-cOMe is likely accounted for at least 40% synaptic translation. CBP20/80, Y14 and eIF4AIII are evidently reduced but eIF4E is increased (Fig. 5). The authors claimed not possible to demonstrate localization defect of a specific target mRNA but the experiment like in-situ hybridization to detect target mRNA instead of Ms2 tethered assay does not require the generation of new transgenic flies. Based on the new Fig 6a data- a dramatic reduction of new protein synthesis, I expect that in-situ signal of poly(A)-positive mRNAs should be significantly decreased in double-KO flies. Without the in-situ hybridization of poly(A)+ RNA, there is no data to demonstrate that "levels of untranslated mRNAs are reduced at synapses in the absence of cOMe" (line 300)? If no specific target mRNA can be

demonstrated in this study, they should at least compare the signal changes of puromycin labeling, poly(A)+ RNA (in situ hybridization), CBP20/80 and eIF4E between synapses and soma in wild-type and dKO flies. I suspect that the changes may also be observed in the soma. If so, the lack of CMTr1 and CMTr2 may just simply affect nuclear export instead of synaptic transport.

The increased levels of eIF4E at synapses shown in Fig 5f indicate a compensatory response to reduced translation in the absence of CMTr's shown in Fig 6a. Accordingly, we did not find a reduction of all mRNA at synapses by in situ hybridization with an oligodT probe, in contrast to the ventral nerve cord where mRNA levels are indeed reduced. Marking untranslated mRNAs with EJC complex components does not allow to quantify the amount of this pool of mRNAs compared to previously translated mRNA from our data. However, our data suggest that there is a large amount of mRNAs present at synapses, but what fraction is translated and what fraction is stored requires a more detailed analysis using individually tagged transcripts expressed from transgenes to analyse localization and translation status in detail.

As detailed in the discussion, our data suggest a model for CMTrs in establishing/maintaining the appropriate repertoire of locally-translated synaptic factors in synapses, that are necessary to support reward learning, rather than directly in learning-induced synaptic changes.

We further added data in Supplementary Fig S8 to show cellular localization of CBP20, eIF4E, puromycin incorporation, which are comparable in wild type and CMTr1/2 double mutants indicating that lack of CMTrs does not simply affect nuclear export. Our data do not indicate major changes in the cell body, but for accurate quantifications, individually tagged transcripts expressed from transgenes need to be analyzed for localization in the cell body and at synapses.

4. The author mentioned "In addition, cOMe is only in the low percent range in mice 9,40 and about 80% in Drosophila (Fig. 1g, h and j, k)9. For a primary role of cOMe for self/non-self discrimination one would expect 100% cOMe (line 209-211)." To our knowledge, N1-cOMe in mammals is detected in > 90 to 100% mRNA in several tissues and cell lines, so it is unclear why the authors said "low% in mice"?

Data published by co-author Rupert Fray in Kruse et al. show that only a fraction of mRNA carries cOMe on the first cap-adjacent nucleotide.

In addition, a recent paper by Lee YL et al. about CMTr1 in mice also shows that only a fraction (about 30% of A and C are Am and Cm) of mRNA carries cOMe on the first cap-adjacent nucleotide (1D TLC, Fig 4C).

Fig 4C in Lee YL et al., Cell Reports 33: 108269

Reviewer #2 (Remarks to the Author):

The authors addressed most of my comments and I congratulate them for the interesting work. There are several exciting questions for follow up studies, in particular regarding the specificity of the enzymes and the potential dynamicity of the modification.

We thank this reviewer for accepting our revision. Indeed, there will be several exciting questions to follow up in this exciting new area of epitranscriptomics.

Reviewer #3 (Remarks to the Author):

The authors have addressed my comments, including doing an experiment that was requested and adding the data to a figure in the paper. Congratulations on a very nice manuscript.

We are pleased that this reviewer accepted our revision.

REVIEWERS' COMMENTS

Reviewer #1 (Remarks to the Author):

I have no more requests or questions regarding experiments. However, 2 mistakes in the text should be fixed.

1) Precisely, “decapping” should be done by yDcps. RppH and TAP are pyrophosphatase, not a decapping enzyme. The removal and radiolabeling of the first 5' phosphate were carried out by Antarctic phosphatase and T4 PNK, respectively. Thus, the statement in lines 102-104 is incorrect, please revise it.

2) The authors said “Data published by co-author Rupert Fray in Kruse et al. show that only a fraction of mRNA carries cOMe on the first cap-adjacent nucleotide. In addition, a recent paper by Lee YL et al. about CMTr1 in mice also shows that only a fraction (about 30% of A and C are Am and Cm) of mRNA carries cOMe on the first cap-adjacent nucleotide (1D TLC, Fig 4C).” Thus, they concluded that cOMe is only about 30% in mice (line 209). This is a total misinterpretation of other people’s data. Because it is difficult to ascertain that radiolabeled nucleotides in the TLC plate are all from the first ribonucleotide of mRNA (due to variations in the amount of contaminated residual rRNA, miRNA and degraded poly(A) RNA fragments) and the limited resolution of the TLC method, the signal ratio of methylated pNTP to unmethylated pNTP does not reflect N1-cOMe efficiency. Even in the authors’ data, by using 2 rounds of poly(A) selection or poly(A) selection with rRNA depletion to highly enrich poly(A) RNA, they also observed radiolabelled NTPs in the sample without decapping treatment (Supp Fig. 2b). Lee et al did not claim that only 30% of N1 capping in the mouse brain based on their 1-D TLC result. Their data only indicated that no other methyltransferases can substitute CMTR1 for N1-cOMe in the brain. In contrast, the studies using isotope labelling and HPLC (Furuichi 1975 PNAS 113:596) and mass-spectrometry CapQuant (Wang 2019, NAR 47:e130) reported ~90-100% N1-cOMe efficiency in mammalian cells and mouse tissues. Both methods quantified N1-cOMe only when the m7G cap was also detected. Thus, I believe their conclusion is more accurate than the percentage calculated from the TLC results. Similarly, based on the TLC results in Fig. 1g, h and j, k, I also have difficulty to understand how the authors can get “80% cOMe” in Drosophila. Please describe how the TLC signals are quantified to get the number.

Point by point response to reviewers' comments

We are very pleased to hear that reviewer 1 has accepted our revisions and thank reviewer 1 for the additional comments on our exciting discoveries.

Reviewer #1 (Remarks to the Author):

1) Precisely, “decapping” should be done by yDcps. RppH and TAP are pyrophosphatase, not a decapping enzyme. The removal and radiolabeling of the first 5' phosphate were carried out by Antarctic phosphatase and T4 PNK, respectively. Thus, the statement in lines 102-104 is incorrect, please revise it.

We now say in the text that RppH is a pyrophosphatase, however, both yDcps and pyrophosphatases remove the cap. Hence, the term “decapping” is used in the field.

2) The authors said “Data published by co-author Rupert Fray in Kruse et al. show that only a fraction of mRNA carries cOMe on the first cap-adjacent nucleotide. In addition, a recent paper by Lee YL et al. about CMTr1 in mice also shows that only a fraction (about 30% of A and C are Am and Cm) of mRNA carries cOMe on the first cap-adjacent nucleotide (1D TLC, Fig 4C).” Thus, they concluded that cOMe is only about 30% in mice (line 209). This is a total misinterpretation of other people’s data. Because it is difficult to ascertain that radiolabeled nucleotides in the TLC plate are all from the first ribonucleotide of mRNA (due to variations in the amount of contaminated residual rRNA, miRNA and degraded poly(A) RNA fragments) and the limited resolution of the TLC method, the signal ratio of methylated pNTP to unmethylated pNTP does not reflect N1-cOMe efficiency. Even in the authors’ data, by using 2 rounds of poly(A) selection or poly(A) selection with rRNA depletion to highly enrich poly(A) RNA, they also observed radiolabelled NTPs in the sample without decapping treatment (Supp Fig. 2b). Lee et al did not claim that only 30% of N1 capping in the mouse brain based on their 1-D TLC result. Their data only indicated that no other methyltransferases can substitute CMTR1 for N1-cOMe in the brain. In contrast, the studies using isotope labelling and HPLC (Furuichi 1975 PNAS 113:596) and mass-spectrometry CapQuant (Wang 2019, NAR 47:e130) reported ~90-100% N1-cOMe efficiency in mammalian cells and mouse tissues. Both methods quantified N1-cOMe only when the m7G cap was also detected. Thus, I believe their conclusion is more accurate than the percentage calculated from the TLC results. Similarly, based on the TLC results in Fig. 1g, h and j, k, I also have difficulty to understand how the authors can get “80% cOMe” in *Drosophila*. Please describe how the TLC signals are quantified to get the number.

We have described now in the text and the methods section how signals were quantified from TLCs and gels. We agree with this reviewer, that the amount of N1-cOMe measured on TLCs is an underestimate even though the frequency of the first transcribed nucleotide is accurately reflected. For this reason we quantified N1-cOMe from the gel shown in Fig 1g. Methylation of the N7-position in the cap guanosine has been viewed as constitutive, but e.g according to PMID: 34125914 it is regulated as well suggesting that the CapQuant approach is an overestimate.